# GLI transcriptional repression regulates tissue-specific enhancer activity in response to Hedgehog signaling

Rachel K Lex[1†], Zhicheng Ji[2†], Kristin N Falkenstein[1†], Weiqiang Zhou[2], Joanna L Henry[1], Hongkai Ji[2], Steven A Vokes[1]*

[1]Department of Molecular Biosciences, The University of Texas at Austin, Austin, United States; [2]Department of Biostatistics, Johns Hopkins Bloomberg School of Public Health, Baltimore, United States

**Abstract** Transcriptional repression needs to be rapidly reversible during embryonic development. This extends to the Hedgehog pathway, which primarily serves to counter GLI repression by processing GLI proteins into transcriptional activators. In investigating the mechanisms underlying GLI repression, we find that a subset of GLI binding regions, termed HH-responsive enhancers, specifically loses acetylation in the absence of HH signaling. These regions are highly enriched around HH target genes and primarily drive HH-specific transcriptional activity in the mouse limb bud. They also retain H3K27ac enrichment in limb buds devoid of GLI activator and repressor, indicating that their activity is primarily regulated by GLI repression. Furthermore, the Polycomb repression complex is not active at most of these regions, suggesting it is not a major mechanism of GLI repression. We propose a model for tissue-specific enhancer activity in which an HDAC-associated GLI repression complex regulates target genes by altering the acetylation status at enhancers.

*For correspondence:
svokes@austin.utexas.edu

†These authors contributed equally to this work

Competing interests: The authors declare that no competing interests exist.

## Introduction

Transcriptional repressors are instrumental in establishing developmental lineages and preventing improper gene expression. Long-term repression is accompanied by stable modifications to DNA and chromatin that prevent rapid transcriptional changes. In contrast, transient repression is rapidly reversible, providing a mechanism for controlling gene activation during the dynamic process of embryogenesis. This control is especially important for spatially restricting gene expression until signal transduction mechanisms alleviate repressor activity. This is exemplified by the Hedgehog (HH) signaling pathway, which ensures proper spatiotemporal regulation of its target genes through the coordination of bifunctional GLI proteins. Activation of HH signaling results in the processing of GLI proteins into transcriptional activators, which are otherwise proteolytically modified into truncated transcriptional repressors in the absence of HH ligand (*Wang et al., 2000*; *Harfe et al., 2004*).

The importance of balancing opposing GLI functions is illustrated in the limb bud, where Sonic Hedgehog (SHH) signaling alleviates GLI repression in a spatiotemporal manner to regulate growth of the digit-forming autopod. HH expression initiates in the posterior, distal limb, and forms a gradient along the posterior-anterior axis. Consequently, GLI activators are enriched in the posterior limb bud where many cells are exposed to HH ligands, while an inverse domain of GLI repressors in the anterior limb bud serve to spatially restrict the boundary of HH target gene expression (*Wang et al., 2000*; *Ahn and Joyner, 2004*). The presence of both GLI activator and GLI repressor domains makes the limb bud an ideal model for understanding the roles of GLI proteins in regulating HH-responsive transcription.

Interestingly, the limb bud is primarily a GLI repressor-driven system, as most transcriptional targets do not actually require GLI activator for transcription, but can be activated by loss of GLI repressor alone. This property of de-repression rather than activation is exemplified by $Shh^{-/-}$ limb buds (constitutive GLI repression, no GLI activation), which have a nearly complete absence of digits and a severe reduction in limb size. The phenotype is markedly improved in $Shh^{-/-};Gli3^{-/-}$ double mutants which lack SHH and the main transcriptional repressor, GLI3, and are therefore devoid of most or all GLI activity (both activation and repression) (*Litingtung et al., 2002*; *te Welscher et al., 2002*; *Bowers et al., 2012*). In particular, GLI de-repression is sufficient to activate most GLI target genes in the limb bud, suggesting that the primary role of the HH pathway is to alleviate GLI repression (*Lewandowski et al., 2015*). The transient nature of GLI repression represents a key mechanism for the dynamic transcriptional regulation of HH targets as HH induction rapidly inactivates GLI repression, resulting in transcription of targets within 4–9 hr of stimulation (*Harfe et al., 2004*; *Panman et al., 2006*; *Visel et al., 2007*).

The mechanisms underlying GLI repression are unknown but could in principle function either by excluding GLI activator binding or by recruiting co-repressors (*Wang et al., 2010*). Although the former category provides an attractive model for how GLI proteins might interpret gradients of HH ligand (*Falkenstein and Vokes, 2014*), it fails to account for the large number of GLI target genes that are fully activated upon de-repression in the absence of HH signaling, and likewise, GLI activator. In support of the latter category, several GLI co-repressors have been identified in various contexts, including Atrophin (*Zhang et al., 2013*), Ski (*Dai et al., 2002*) and tissue-specific transcription factors (*Oosterveen et al., 2012*; *Hayashi et al., 2016*). Members of the BAF chromatin remodeling complex have also been shown to generally regulate GLI transcriptional responses but it is unclear if they specifically regulate GLI repression (*Jagani et al., 2010*; *Zhan et al., 2011*; *Jeon and Seong, 2016*; *Shi et al., 2016*). Additional studies have described various interactions between Polycomb repression and HH signaling (*Wyngaarden et al., 2011*; *Shi et al., 2014*; *Weiner et al., 2016*; *Deimling et al., 2018*), indicating the possibility that PRC2 mediates aspects of GLI repression. Since mutations in candidate repressor complexes are pleiotropic, it has been challenging to determine if they directly mediate GLI repression, a challenge compounded by the dual roles of GLI proteins as transcriptional activators and repressors.

Using a genomic approach and the developing limb as a model, we sought to determine if GLI proteins repress HH target genes through altering the chromatin environment at GLI binding regions (GBRs). We hypothesized that GLI repressors regulate gene expression by inactivating enhancers. Consistent with this, we find that GLI repression regulates enhancer modification status, and thus, activity through the de-acetylation of Histone H3K27. This repression occurs independently of Polycomb activity. Enhancers regulated in this fashion correspond to known GLI limb enhancers, are highly enriched around HH target genes, and primarily drive tissue-specific enhancer activity within HH-specific expression domains. Based on these findings, we propose that GLI repressors inhibit gene expression by altering enhancer activity, providing an explanation for the labile nature of GLI repression.

## Results

### A subset of GLI binding regions is epigenetically regulated by HH signaling

Since most HH targets can be activated by loss of GLI repression, we hypothesized that enhancers may be activated by HH signaling when GLI repression is relieved. To test this, we first identified active GLI enhancers in the developing limb at embryonic day 10.5 (E10.5), when high levels of HH target gene expression are observed. We used an endogenously FLAG tagged *Gli3* allele (*Lopez-Rios et al., 2014*; *Lorberbaum et al., 2016*) to identify GLI3 binding regions by ChIP-seq and then identified regions enriched for H3K27ac, a marker associated with active enhancers (*Heintzman et al., 2007*; *Heintzman et al., 2009*; *Creyghton et al., 2010*; *Rada-Iglesias et al., 2011*; *Cotney et al., 2012*). Altogether we identified 7,282 endogenous GLI3 binding regions (GBRs), with the majority of regions enriched for H3K27ac (83%; 6,064/7,282 GBRs) in wild-type (WT) limb buds which have active HH signaling (*Figure 1—figure supplement 1A*; *Figure 1—source data 1*). Nearly all nuclear GLI3 is present in the anterior half of the limb bud in the repressor form

with little or no nuclear GLI3 present in the posterior half, consistent with previous findings (*Wang et al., 2000*) (*Figure 1—source data 1B*). Therefore, the GBRs identified in this study are likely to exclusively represent GLI3-repressor binding regions.

Next, we asked if HH signaling was required for the activation of GLI enhancers by performing ChIP-seq for H3K27ac in *Sonic hedgehog* (*Shh*) null E10.5 forelimbs, prior to overt phenotypes in *Shh* nulls, and comparing H3K27ac enrichment to that in WT limbs (*Chiang et al., 2001*) (*Figure 1A*; *Figure 1—source data 2*). Since *Shh*$^{-/-}$ forelimbs have constitutive GLI repression, we hypothesized that in the absence of HH signaling, GLI repressors may prevent activation of their enhancers. We found that most H3K27ac enriched regions were present in both WT and *Shh*$^{-/-}$ embryos (98.3%; 58,720/59,729 H3K27ac peaks); however a subset of 2,113 WT H3K27ac enriched regions had acetylation that was significantly reduced or completely lost in the absence of HH signaling (*Figure 1—source data 2*). We then asked whether those regions with reduced acetylation in the absence of HH signaling include GLI-bound enhancers by intersecting H3K27ac enrichment with the endogenous GBRs identified. We found that 94% of GBRs (5,715/6064 GBRs) with acetylation in WT limbs also retain H3K27ac in *Shh*$^{-/-}$ limb buds, which we have termed Stable GBRs (*Figure 1C,D*). GBRs that remain stably acetylated regardless of HH signaling likely function as active enhancers whose activity is not predominantly regulated by HH signaling. However, H3K27ac enrichment was reduced or lost in the absence of HH signaling in a smaller subset of GBRs, suggesting that GLI repressor may regulate the activity of this group of enhancers. Within this GBR class with HH-responsive acetylation, we identified populations of GBRs that had either significant reductions (termed HH-sensitive; n = 148) or a complete absence of H3K27ac enrichment (termed HH-dependent; n = 201) in *Shh*$^{-/-}$ limb buds (*Figure 1B,C*). The latter two categories are henceforth collectively referred to as HH-responsive GBRs. As H3K27ac is not exclusively localized to enhancers, we also examined the enrichment of histone H3K4me1, a general marker of primed and active enhancers, at these GBRs using publicly available data (*ENCODE Project Consortium, 2012*) (*Figure 1—source data 3*). In WT limb buds, 82% of HH-responsive GBRs are enriched for H3K4 mono-methylation, supporting that these regions are likely to act as enhancers (HH-sens: 123/148, 83%; HH-dep: 162/201, 81%).

## Hedgehog-responsive GBRs are enriched near Hedgehog target genes

To determine if HH-responsive GBRs are associated with HH target genes, we examined biologically validated GLI enhancers in the *Gremlin* and *Ptch1* loci that mediate limb-specific transcription of these HH targets and found that they are among the HH-responsive class of GBRs (*Figure 1E,F*) (*Vokes et al., 2008*; *Zuniga et al., 2012*; *Li et al., 2014*; *Lopez-Rios et al., 2014*). This suggests that HH-responsive enhancers may regulate limb-specific gene expression in response to HH signaling. Consistent with this possibility, we found that HH-responsive GBRs are highly enriched around the TSS (2 kb upstream to 1 kb downstream) of genes that have reduced expression in *Shh*$^{-/-}$ limb buds (*Lewandowski et al., 2015*). In contrast, Stable GBRs have minimal, albeit still significant enrichment around HH target genes (p=0, permutation test; *Figure 1G*).

We observed many HH-responsive H3K27ac regions that change acetylation status in response to HH signaling but are not bound by GLI3. This prompted us to ask asked if these regions cluster near GBRs. HH-responsive non-GLI binding regions cluster together and are significantly enriched around HH-responsive GBRs, and to a lesser extent, near Stable GBRs (*Figure 1—figure supplement 1C, D*). We conclude that HH-responsive GBRs cluster with other HH-responsive regulatory regions, and are strongly associated with HH target genes, supporting their role in driving gene expression in response to HH signaling during limb development.

## HH-responsive GBRs are distal enhancers containing high quality GLI motifs

Although Stable GBRs are not highly enriched at HH target genes, 62% of them (3,544/5,715) are located in close proximity to the promoters of genes (2 kb upstream to 1 kb downstream of TSS), compared to 26% (91/349) of HH-responsive GBRs (*Figure 1H*). Most promoter-associated Stable GBRs (90%; 3,190/3,544) are found at promoters associated with CpG islands (defined as a TSS with a CpG region within 5 kb upstream to 2.5 kb downstream), a quality typically associated with housekeeping genes, and genes that tend to be more broadly expressed and less tissue-specific (*Zhu et al., 2008*). To examine how different classes of GBRs might be differentially regulated, we

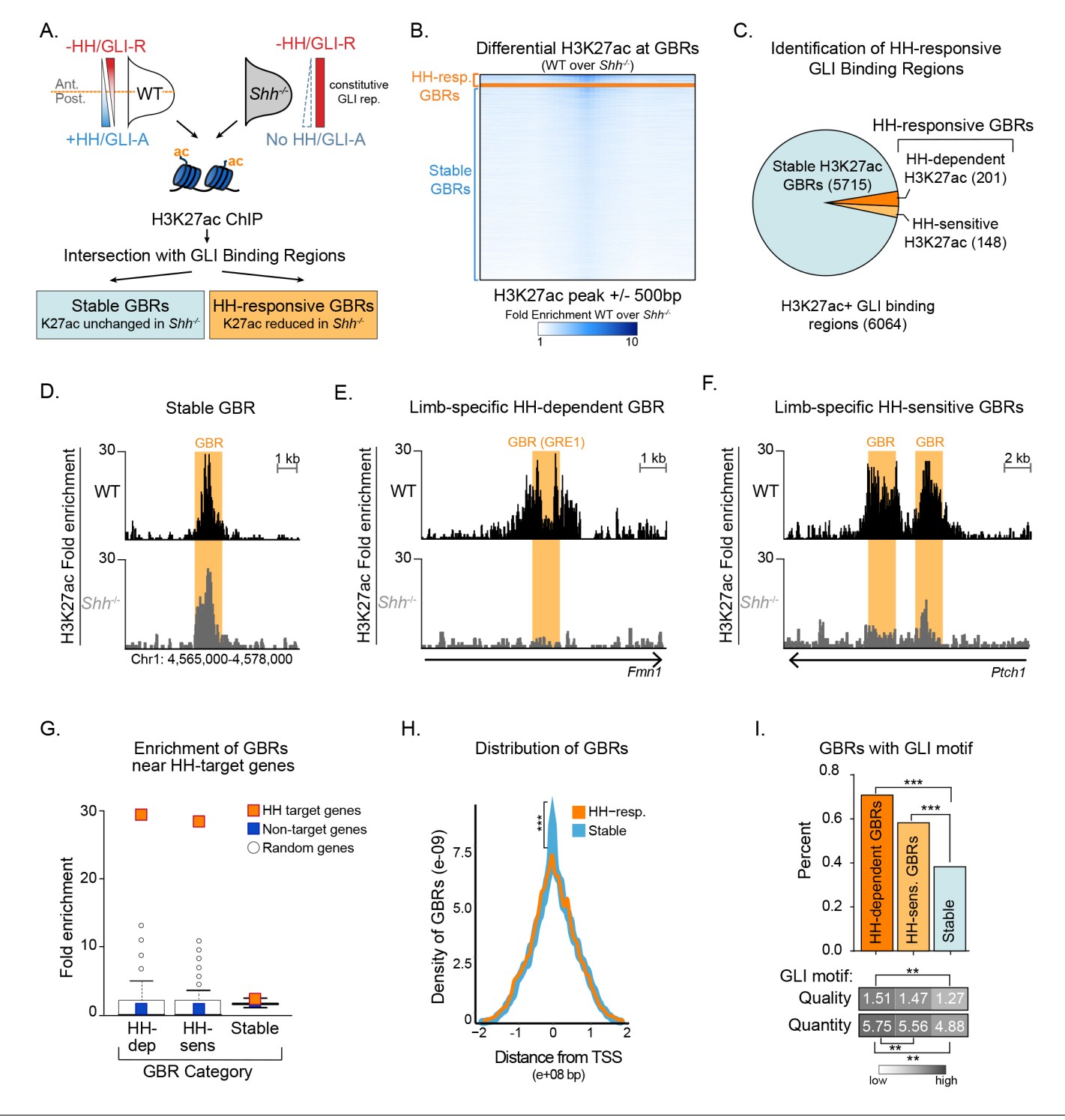

**Figure 1.** Hedgehog signaling regulates acetylation of H3K27 at a subset of GLI binding regions. (**A**) Pipeline for identifying different categories of GLI bound regions (GBRs). (**B**) Heatmap depicting differential H3K27ac enrichment in WT over *Shh*$^{-/-}$ limb buds for HH-responsive and Stable GBRs. (**C**) Classification of GBR categories from E10.5 GBRs with H3K27ac in WT limbs. (**D-F**). H3K27ac enrichment in WT and *Shh* $^{-/-}$ is shown across a representative genomic region near a Stable GBR (**D**), and biologically validated HH-responsive GBRs: a HH-dependent GBRs, GRE1, at the HH target gene *Gremlin 1* (*Grem1*) (*Li et al., 2014*) (**E**) and HH-sensitive GBRs shown to regulate limb-specific expression of the HH target *Ptch1* (*Lopez-Rios et al., 2014*) (**F**). (**G**) HH-dependent GBRs, HH-responsive GBRs and Stable GBRs are significantly enriched (2 kb upstream- 1 kb downstream of TSS) near HH target genes compared to randomly chosen genes (p=0, p=0 and p=0, respectively, permutation test based on 1000 permutations). (**H**) *Figure 1 continued on next page*

*Figure 1 continued*

Proportional distribution of Stable and HH-responsive GBRs arounds transcription start sites (TSS), indicating significant enrichment of Stable GBRs at TSS compared to HH-responsive GBRs (p=2.55e-40, Fisher's exact test, two sided). (I) Both HH-dependent and HH-sensitive GBRs have significantly more GLI motifs than Stable GBRs (top)(p=2.2e-16 and p=8.00e-06; one-sided proportional test). HH-dependent and HH-sensitive GBRs containing GLI motifs have significantly higher quality of GLI motifs than Stable GBRs (Quality score; p=5.03e-13 and p=5.98e-08; one-sided Wilcoxon test) and significantly more motifs per GBR within HH-dependent GBRs than Stable GBRs (Quantity score; p=5.92e-06; one-sided Wilcoxon test). See *Figure 1—figure supplement 1*, *Figure 1—source data 1*, *Figure 1—source data 2*, *Figure 1—source data 3*, *Figure 1—source data 4*.

The online version of this article includes the following source data and figure supplement(s) for figure 1:

**Source data 1.** Endogenous GLI3-Flag ChIP-seq analyzed data and called peaks.
**Source data 2.** WT vs *Shh*^-/- H3K27ac ChIP-seq analyzed data and called peaks.
**Source data 3.** H3K4me1 ChIP-seq analyzed data and called peaks from GSE86690.
**Source data 4.** Motifs uncovered from HH-responsive enhancers.
**Figure supplement 1.** Nuclear localization of GLI3 and properties of GLI binding regions.

examined their GLI binding motifs. A significantly higher percentage of HH-dependent and HH-sensitive GBRs contain GLI motifs compared to Stable GBRs (69.7% HH-dep., 57.4% HH-sens., 39.5% Stable). HH-dependent and HH-sensitive GBRs also contain a higher density (1.51 HH-dep., 1.47 HH-sens., 1.27 Stable) and higher quality of GLI motifs compared to Stable GBRs (5.75 HH-dep., 5.56 HH-sens., 4.88 Stable)(*Figure 1I*). Interestingly, we did not uncover high levels of enrichment of other motifs using de novo motif analysis (*Figure 1—source data 4*). Additionally, Stable GBRs are slightly more conserved than HH-dependent, but not HH-sensitive GBRs (see Discussion) (*Figure 1—figure supplement 1E*).

## The Polycomb repressor complex does not regulate most GLI enhancers

GLI activators have been shown to recruit demethylases that remove H3K27me3, a hallmark of the Polycomb repressor complex (PRC2) to promote transcriptional activation of several HH target genes, most notably *Gli1* and *Ptch1* (*Margueron and Reinberg, 2011*; *Shi et al., 2014*; *Lorberbaum et al., 2016*). If PRC2 is recruited by GLI repressors, there should be enrichment of H3K27me3 at HH-responsive enhancers in *Shh*^-/-, where maximal levels of GLI repression would lead to recruitment of PRC2 and thus methylation at these enhancers. Contrary to this prediction, we identified a minimal number of HH-responsive GBRs enriched for H3K27me3 in E10.5 *Shh*^-/- limb buds (31/349 GBRs; *Figure 2A–C*, *Figure 2—source data 1*). As reported for MEFs (*Shi et al., 2014*), these methylated GBRs include the pathway target *Gli1* in addition to other pathway target genes such as *Ptch1* and *Ptch2* (*Figure 2B*). In contrast, most HH-responsive GBRs (318/349) and signature target gene promoters (14/22) lack enrichment of H3K27me3 in the absence of HH signaling (*Figure 2C*; *Figure 2—figure supplement 1*; *Figure 2—source datas 1* and *2* ). We conclude that while the PRC2 complex has the potential to regulate a small number of HH pathway target genes, it is not the primary mechanism by which GLI repressors prevent target gene expression.

## Hedgehog signaling does not regulate other histone modifications at enhancers

We considered two possible mechanisms by which GLI repression could regulate H3K27ac enrichment in response to HH signaling: first, GLI repression could cause large-scale modifications to chromatin at enhancers resulting in an overall loss of their identity as enhancers. Alternatively, GLI repressors could regulate H3K27ac specifically. To address the first mechanism, we asked if HH regulates H3K4me2, another histone modification enriched at active enhancers and most promoters (*Ernst et al., 2011*; *Pekowska et al., 2011*; *Wang et al., 2014*). Consistent with H3K4me2 being enriched at promoters and our finding that Stable GBRs are enriched around promoters, we find H3K4 di-methylation at 73% of Stable GBRs (4,172/5,715), while only 26% of HH-responsive GBRs (91/349) which are less enriched around promoters. None of the GLI-bound H3K4me2 enriched regions had significant reductions in H3K4me2 in *Shh*^-/- limbs compared to WT controls (*Figure 2D, E*). Furthermore, essentially all peaks remained unchanged between the two genotypes, where only 12 peaks were reduced in *Shh*^-/- limbs, none overlapping with GLI binding regions or non-GBR HH-responsive peaks (*Figure 2—source data 3*).

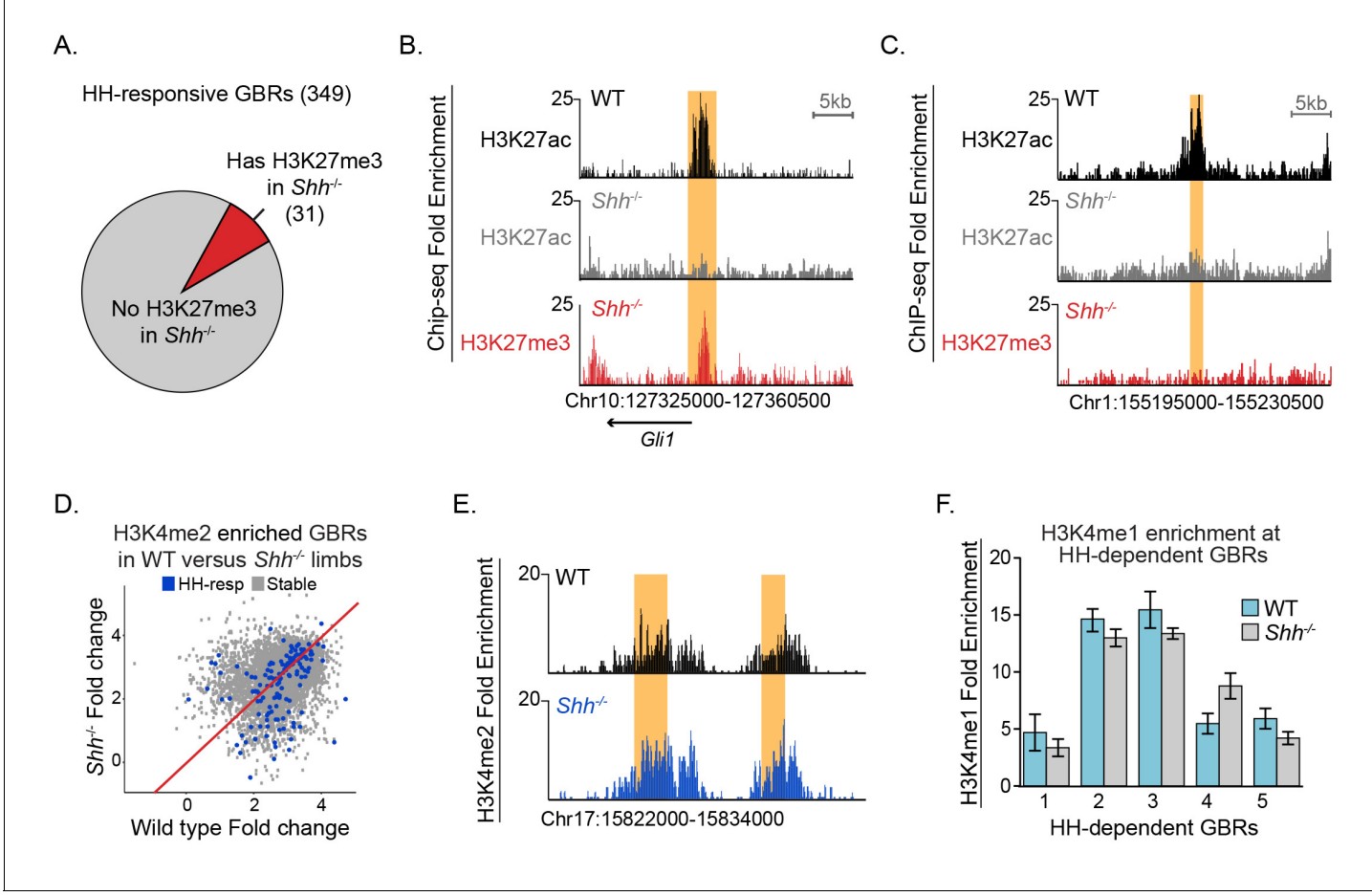

**Figure 2.** Most HH-responsive GBRs are not regulated by Polycomb repression and retain markers of poised enhancers. (A) Chart depicts HH-responsive GBRs that contain enrichment for the PRC2 marker H3K27me3 in *Shh*^-/- limb buds (n = 2). (B) Tracks depicting a HH-responsive region in *Gli1* with differential H3K27ac enrichment in WT and *Shh*^-/- limb buds and H3K27me3 enrichment in *Shh*^-/- limb buds. (C) Tracks depicting a representative HH-dependent GBR that also lacks H3K27me3. (D) Scatter plot for H3K4me2 enrichment of Stable and HH-responsive GBRs from WT and *Shh*^-/- limb buds (n = 2). No GBRs show significant changes in di-methylation of H3K4 between WT and *Shh*^-/-. (E) Representative track showing comparable levels of H3K4me2 enrichment for a HH-responsive GBR in WT and *Shh*^-/- limb buds. (F) Quantitative-PCR assays indicating H3K4me1 ChIP enrichment in WT and Shh-/- limb buds at HH-dependent GBRs (n = 2). See *Figure 2—figure supplement 1*, *Figure 2—source data 1*, *Figure 2—source data 2*, *Figure 2—source data 3*.

The online version of this article includes the following source data and figure supplement(s) for figure 2:

**Source data 1.** *Shh*^-/- H3K27me3 ChIP-seq analyzed data and called peaks.
**Source data 2.** Hedgehog responsive genes with H3K27me3 enrichment.
**Source data 3.** WT vs *Shh*^-/- H3K4me2 ChIP-seq analyzed data and called peaks.
**Figure supplement 1.** H3K27Me3 enrichment at the promoters of GLI target genes.

H3K4me2 marked most Stable GBRs, but only a subset of HH-responsive GBRs which are primarily located within 2 kb upstream to 1 kb downstream of TSS (79% (72/91) of H3K4me2+ HH-responsive GBRs are near promoters). Since we found that most HH-responsive GBRs in wildtype limb buds are enriched for H3K4me1 (see results above), we asked if this mark was altered at HH-responsive GBRs in response to HH signaling. We performed ChIP on WT and *Shh*^-/- limb buds and assessed enrichment of H3K4me1 at several HH-responsive GBRs by quantitative PCR, selecting intergenic regions that would not overlap with promoters (regions are at least 7 kb from the nearest TSS). All tested regions retained H3K4me1 enrichment in *Shh*^-/- limb buds (*Figure 2F*). We conclude that HH-responsive regions retain enrichment of other active or poised enhancer marks, suggesting that HH signaling and GLI repression specifically regulate H3K27ac enrichment at these regions.

## Chromatin at HH-responsive GBRs compacts in the absence of Hedgehog

The dynamic acetylation of HH-responsive GBRs, yet unaltered methylation of H3K4 in *Shh*[-/-] limb buds are properties consistent with 'poised' enhancers, which retain H3K4me1 and accessible chromatin in the absence of H3K27ac (*Heintzman et al., 2009*; *Creyghton et al., 2010*; *Rada-Iglesias et al., 2011*). Therefore, if HH-responsive enhancers are not active but 'poised' in the absence of HH, we predicted that chromatin accessibility would be unchanged in response to HH signaling. Using ATAC-seq to measure regions of open chromatin, we compared the accessibility of GBRs between WT and *Shh*[-/-] posterior limb buds, a fraction providing a more homogenous WT population of cells exposed to HH signaling (*Figure 3A*; *Figure 3—source data 1*) (*Buenrostro et al., 2013*; *Buenrostro et al., 2015*). Overall, in HH stimulated WT limbs, 87% of

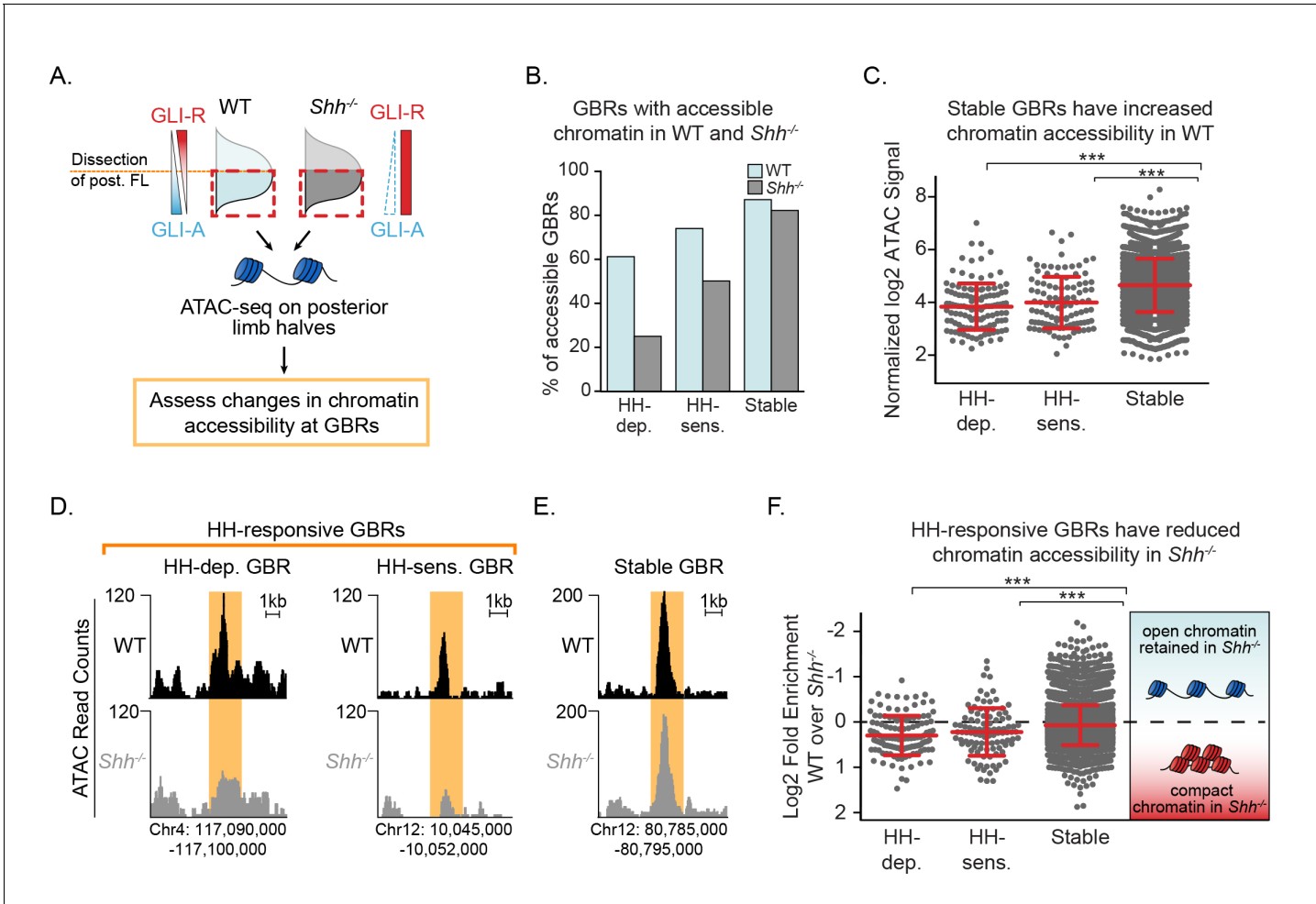

**Figure 3.** Chromatin accessibility is reduced in the absence of Hedgehog signaling. (A) ATAC-seq pipeline for single pairs of dissected posterior halves of forelimbs (n = 2). ATAC peaks, signifying accessible chromatin regions were intersected with Stable GBRs and HH-responsive GBRs. (B) Many HH-responsive GBRs that are accessible in WT limb buds are inaccessible *Shh*[-/-] limb buds, while the accessibility of Stable GBRs remains largely unchanged. (C) Plot of log2 normalized signal in chromatin accessibility in WT limbs indicating that Stable GBRs are more accessible than HH-dependent and HH-responsive GBRs (p=3.98e-19, p=9.21e-11; Wilcoxon rank sum test). Each data point represents a single GBR and red bars indicate the median, upper and lower quartiles. D-E. Representative ATAC-seq peaks showing lack of accessibility in *Shh*[-/-] limb buds at HH-responsive GBRs (D), but not in Stable GBRs (E, F) Plot of log2 fold changes in chromatin accessibility in the presence and absence of HH signaling. HH-responsive GBRs are significantly less accessible than Stable GBRs (Stable vs. HH-sensitive. p=0.001; Stable vs. HH-dependent p=4.99e-09; Wilcoxon rank sum test). See *Figure 3—source data 1*.

The online version of this article includes the following source data for figure 3:

**Source data 1.** WT vs *Shh*[-/-] ATAC Seq analyzed data and called peaks.

Stable GBRs (4,978/5,715) are accessible, while only 66% of HH-responsive GBRs (232/349) are accessible, suggesting a more restricted accessibility of HH-responsive GBRs even in WT conditions (*Figure 3B–C*). To determine if these regions are likely to be enhancers, we analyzed the co-enrichment of the enhancer markers H3K4me1 and H3K4me2 at ATAC accessible (ATAC+) and inaccessible (ATAC-) HH-responsive GBRs. 93.5% (217/232) of ATAC+ regions are co-enriched with H3K4me1/2 while 72% (84/117) of ATAC- regions are co-enriched with H3K4Me1/2. These results suggest that most of the ATAC- regions are likely to correspond to real enhancers though at a somewhat reduced frequency compared to ATAC+ regions. Contrary to expectations for a poised enhancer, both HH-sensitive and HH-dependent GBRs have significantly reduced accessibility compared to Stable GBRs in the absence of HH signaling, with the majority of HH-responsive GBRs being more compact in *Shh*$^{-/-}$ compared to wild-type limbs (*Figure 3D–F*). Overall, we conclude that HH-responsive GBRs are less accessible than Stable GBRs, with access being further restricted in *Shh*$^{-/-}$ limb buds, which have constitutive GLI repression.

## De-repression is the dominant mechanism regulating GLI enhancer activation

The presence of multiple GLI proteins and their bifunctional roles as both transcriptional activators and repressors has made it challenging to determine how HH genes are primarily regulated. To test the roles of activator and repressor on enhancers, we performed H3K27ac ChIP on *Shh*$^{-/-}$;*Gli3*$^{-/-}$ limb buds (devoid of GLI activators and most GLI repressors). We hypothesized that loss of H3K27ac at most HH-responsive enhancers in the absence of HH signaling is due to constitutive GLI repression preventing acetylation of GLI enhancers. Thus, in *Shh*$^{-/-}$;*Gli3*$^{-/-}$ limbs, we predicted H3K27ac should be maintained at HH-responsive enhancers. Alternatively, if GLI activator is required, H3K27ac would remain absent or reduced as it does in *Shh*$^{-/-}$ limbs (*Figure 1A*).

To overcome the reduced tissue available for ChIP samples, we optimized a 'MicroChIP' approach to allow ChIP-seq on single pairs of limb buds and assessed H3K27ac enrichment at GLI enhancers in E10.5 *Shh*$^{-/-}$;*Gli3*$^{-/-}$ limb buds (*Figure 4A*; *Figure 4—source data 1*). As anticipated, there was reduced signal compared to our standard protocol, however we were still able to detect many of the HH-responsive GBRs (59%; 207/349) and most Stable GBRs (91%; 5,211/5,715). Consistent with expectations, HH-responsive GBRs associated with *Gli1* and *Ptch1*, which require GLI activation (*Litingtung et al., 2002*; *te Welscher et al., 2002*), had greatly reduced H3K27ac enrichment in the double mutants along with a small number of additional GBRs (24 total; *Figure 4B,D,E*). However, consistent with a GLI repression-driven model, most HH-responsive GBRs retained or increased H3K27ac enrichment in the absence of both GLI activator and repressor (88%; 183/207; *Figure 4C–E*). Despite being unchanged in *Shh*$^{-/-}$ limbs, Stable GBRs had slight but significant increases in H3K27ac enrichment (*Figure 4F*), indicating that on a population level, some of these regions respond to GLI repression (see Discussion).

In a parallel series of experiments, we noted that HH-responsive GBRs have higher levels of H3K27ac enrichment in posterior limb halves, where HH is active, compared to anterior limb halves, which have little exposure to HH and are dominated by GLI repression (*Figure 4G*). This contrasts with *Gli3*$^{-/-}$ limb buds, where H3K27ac levels in anterior halves are comparable to those in posterior halves in many GBRs, as both domains lack GLI repression (*Figure 4H*). Together these results strongly support a GLI repressor centric mode of regulation where GLI de-repression is responsible for activation of most GLI limb enhancers. We conclude that GLI activator does not mediate acetylation levels at most HH-responsive GBRs.

## HDACs dynamically regulate H327ac enrichment at HH-responsive enhancers

The simplest interpretation of the above results is that GLI repressor regulates the activity of histone deacetylases (HDACs) at HH-responsive GBRs, in which loss of an HDAC-GLI repressor complex leads to acetylation. To test this, we cultured limb buds in the presence of the HDAC inhibitors FK228 (*Furumai et al., 2002*) or SAHA for 2 hr. As expected, there were greatly upregulated levels of H3K27ac within two hours of treatment (*Figure 5—figure supplement 1A,B*). We then dissected the anterior halves of limb buds cultured in control or HDAC inhibitor-containing media and compared the levels of H3K27ac enrichment at HH-responsive GBRs previously shown to have enriched

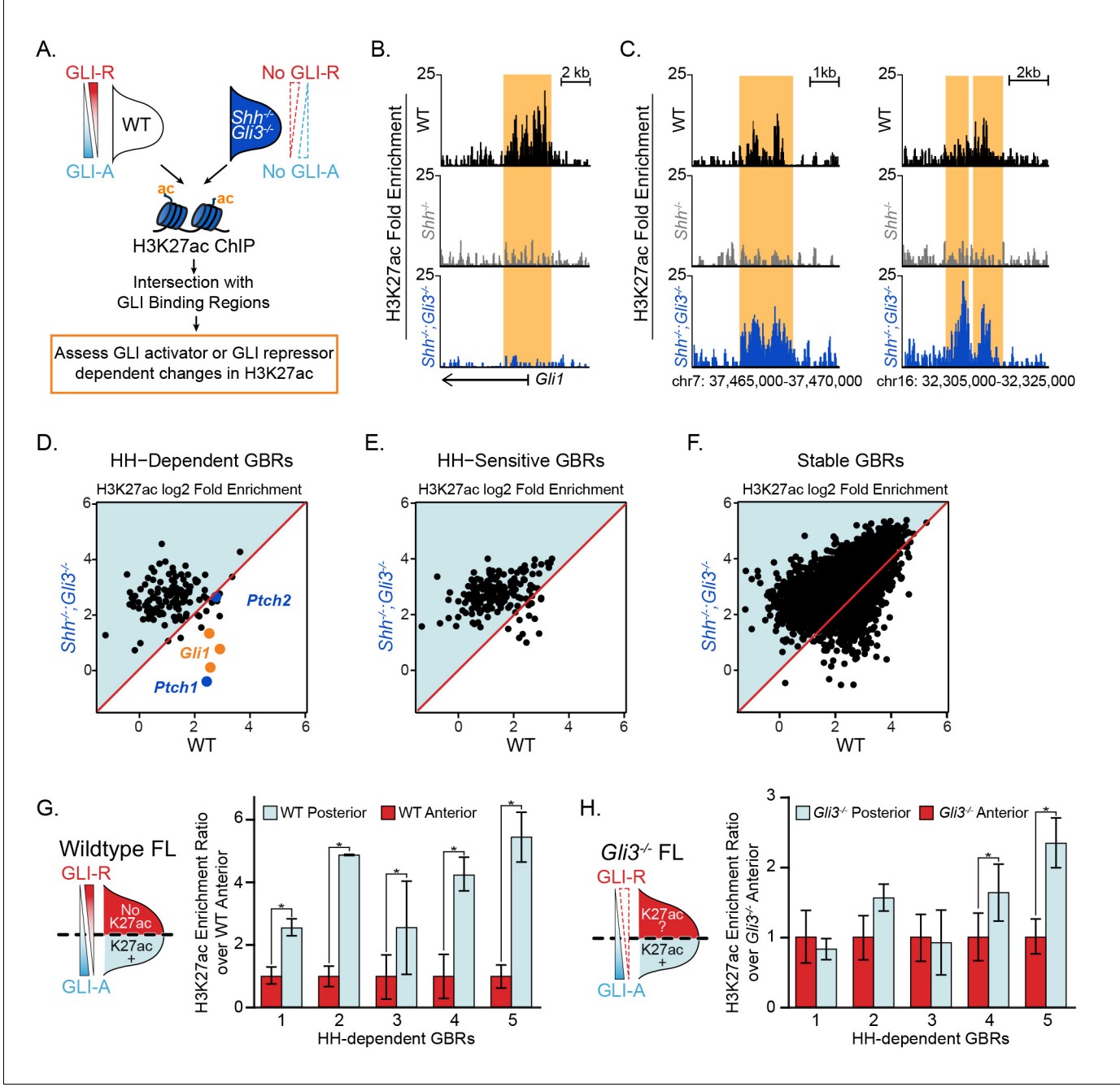

**Figure 4.** GLI de-repression activates most HH-responsive enhancers. (A) *Shh⁻/⁻;Gli3⁻/⁻* H3K27ac 'MicroChIPs' on single pairs of E10.5 forelimbs (33–34S) *Shh⁻/⁻;Gli3⁻/⁻* and WT littermate controls (n = 2, respectively). (B) A HH-responsive GBR near *Gli1* which requires GLI activator for H3K27ac enrichment. (C) Representative examples of HH-responsive GBRs, activated by loss of GLI repressor that do not require GLI activator. (D-F) Scatter plot of H3K27ac enrichment of HH-dependent, HH-sensitive and Stable GBRs in WT and *Shh⁻/⁻;Gli3⁻/⁻* limbs. Each dot represents a single GBR. The p-values indicate a significant enrichment of acetylation in *Shh⁻/⁻;Gli3⁻/⁻* among all GBR classes (p-values: HH-dependent = 2.26e-08, HH-sensitive = 5.41e-11, Stable = 3.4e-185;Wilcoxon-rank sum tests). (G-H) E10.5 WT and *Gli3⁻/⁻* limb buds were dissected into anterior and posterior halves as indicated and selected HH-dependent GBRs were tested for H3K27ac enrichment by quantitative PCR in each fraction (n = 4). HH-dependent GBRs have higher ratios of posterior to anterior H3K27ac enrichment in WT limb buds (G), while many HH-dependent GBRs have equal ratios of posterior to anterior H3K27ac enrichment in *Gli3⁻/⁻* limb buds (H) (n = 3) (asterisks indicate p<0.05, paired T-test). See *Figure 4—source data 1*.

The online version of this article includes the following source data for figure 4:

**Source data 1.** WT vs *Shh⁻/⁻;Gli3⁻/⁻* H3K27ac MicroChIP-seq analyzed data and called peaks.
**Source data 2.** MicroChIP H3K27ac enrichment in *Shh⁻/⁻;Gli3⁻/⁻* limb buds at HH-responsive GBRs with H3K27me3 in *Shh⁻/⁻* limbs.

H3K27ac levels in posterior limb halves (+HH, no GLI repression) (*Figure 4G*). Inhibition of HDACs with both FK228 and SAHA resulted in increased acetylation at HH-responsive enhancers compared to untreated control anterior limb buds (*Figure 5A*). The increased enrichment of H3K27ac acetylation in HDAC-inhibited anterior limb buds was comparable to that seen in posterior limb buds (*Figure 4G*). HDACs could regulate H3K27ac activity in a GLI-responsive fashion through a variety of different mechanisms including direct interactions with responsive GBRs, potentially mediated by a repression complex including GLI3 proteins and HDACs. We asked if GBRs were bound by HDACs, focusing on HDAC1, which along with HDAC2 is preferentially inhibited by FK228 (*Furumai et al., 2002*). We identified HDAC1 binding regions in E11.5 limb buds by CHIP-seq and intersected them with GBRs. 78% (4,109/5,282) of stable GBRs and 41% (144/349) of HH-responsive GBRs overlapped with HDAC1 peaks (*Figure 5B–D*), consistent with a possible role for HDAC1 in regulating H3K27ac levels. We conclude that GLI repressors regulate H3K27ac levels at HH-responsive GBRs through HDACs (see discussion).

## HH-responsive GBRs have increased tissue-specificity compared to Stable GBRs

Having identified distinct classes of GBRs that respond differently to HH signaling, we next addressed the biological significance of these properties. To this end, we used the VISTA enhancer database to identify a total of 305 Stable and 23 HH-responsive GBRs that had been tested for enhancer activity in transgenic embryos (*Visel et al., 2007*). While nearly half of each class have

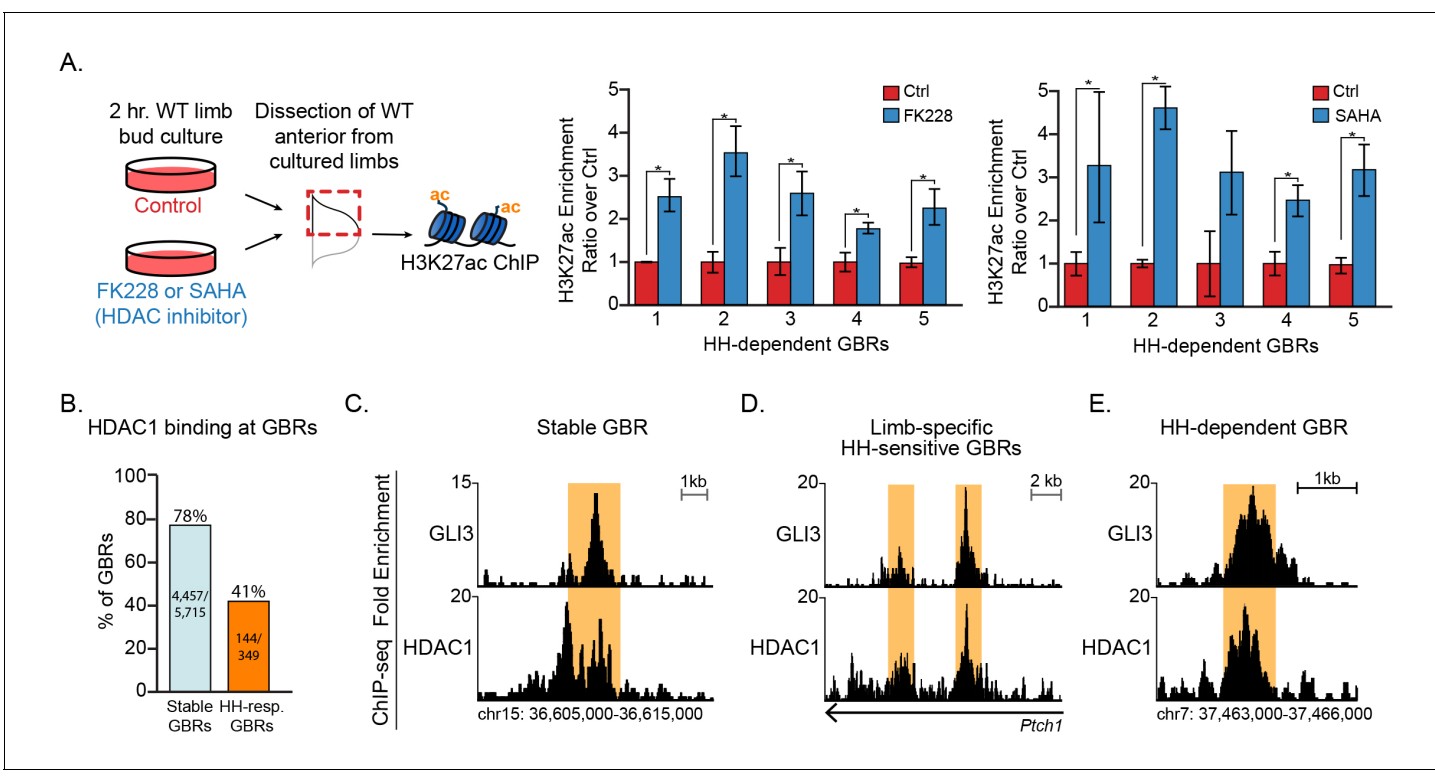

**Figure 5.** HDACs regulate H3K27ac at HH-responsive GBRs. (A) Inhibition of HDACs using 250 nM of FK228 or 20 μM SAHA in cultured limb buds for two hours resulted in significant increases of H3K27ac enrichment at HH-dependent GBRs from anterior cultured limb buds compared to DMSO control anterior limbs (FK228 n = 4; SAHA n = 5; asterisks indicate p<0.05, paired T-test). (B) HDAC1 binding at Stable and HH-responsive GBRs (n = 4). (C-E) HDAC1 at GLI3 binding regions, shown at a representative Stable GBR (C), limb-specific HH-sensitive GBRs near the HH target genes *Ptch12* (*Lopez-Rios et al., 2014*) (D), and a HH-dependent GBR, (region also shown in *Figure 4C*) (E, D) See *Figure 5—figure supplement 1*, *Figure 5—source data 1*.

The online version of this article includes the following source data and figure supplement(s) for figure 5:

**Source data 1.** HDAC1 ChIP-seq analyzed data and called peaks.
**Figure supplement 1.** H3K27ac is increased upon HDAC inhibition.

enhancer activity in the limb, HH-responsive GBRs tend to drive activity specific to the HH-responsive posterior limb bud, while Stable GBRs tend to have activity throughout the limb or regions that are not responsive to HH (*Figure 6A,B*) (*Ahn and Joyner, 2004*; *Probst et al., 2011*; *Lewandowski et al., 2015*). Additionally, HH-responsive enhancers are active more specifically within the limb (drive expression in an average of 1.9 tissues) while Stable GBRs are more broadly active throughout the embryo (drive expression in an average of 2.9 tissues; p<0.01; *Figure 6C*; *Figure 6—source data 1*). While all GBRs examined in the VISTA database with limb activity are by definition enriched for H3K27ac, 91% of HH-responsive GBRs and 95% of Stable GBRs are also enriched for H3K4me1. Additionally, all GBRS are enriched for at least two markers of enhancers (H3K27ac, H3K4me1, H3K4me2, ATAC) while most are enriched for 3–4 of these markers (67% HH-responsive GBRs; 93% Stable GBRs) (*Figure 6D,E*).

These results suggest that Stable GBRs act as general enhancers that drive expression in multiple tissues, while HH-responsive GBRs mediate tissue-specific expression. To test this in another biological context, we treated HH-responsive NIH3T3 cells with and without the HH agonist purmorphamine, identified H3K27ac enriched regions by ChIP-Seq, and assessed the H3K27 acetylation status of different classes of limb GBRs. Strikingly, only 12% (42/349 GBRs) of HH-responsive limb GBRs are acetylated in response to HH signaling in NIH3T3 cells. An additional 18% (63/349 GBRs) of HH-responsive limb enhancers have stable acetylation in NIH3T3 cells, while most lack any activity. In contrast, 70% (4,001/5715) of Stable GBRs in the limb are still active in NIH3T3 cells in both untreated and HH stimulated cells (*Figure 6F,G*; *Figure 6—source data 2*). We conclude HH-responsive GBRs are tissue specific enhancers that mediate HH signaling, while Stable GBRs have broadly expressed enhancer activity.

## Discussion

We find that a subset of GLI-bound regions have chromatin modifications that change in response to HH signaling. These regions are enriched for multiple enhancer markers and have enhancer activity in transgenic embryos, suggesting that they mark a population of enhancers. However, compared to WT embryos, these regions have reduced or absent levels of histone H3K27 acetylation in *Shh*[-/-] embryos, indicating a loss of enhancer activity. Many previously validated GLI limb enhancers have HH-responsive H3K27ac, including those regulating *Grem1*, *Ptch1* and *Gli1* (*Figure 1E,F*) (*Vokes et al., 2008*; *Zuniga et al., 2012*; *Li et al., 2014*; *Lopez-Rios et al., 2014*). Moreover, HH-responsive GBRs are highly enriched near HH target genes while the much larger class of Stable GBRs are not (*Figure 1G*). This suggests that HH target gene regulation is primarily mediated through HH-responsive GBRs. The discovery of this response provides important information about the mechanism of GLI repression. It also provides a predictive tool for identifying enhancers regulating HH target genes in other biological contexts.

We propose a model in which GLI repression primarily regulates enhancer activity through deacetylation of histone H3K27. Because H3K4me1 and H3K4me2 levels are unchanged during maximal GLI repression, these enhancers presumably remain poised for activation, albeit in a less accessible state. Upon binding HH-responsive enhancers, GLI repressors either recruit or activate HDACs, which prevent otherwise competent enhancers from acquiring enriched H3K27 acetylation. The loss of GLI repression, either genetically (*Shh*[-/-]*;Gli3*[-/-] or *Gli3*[-/-] limb buds), or developmentally (initiation of *Shh* expression) results in a loss of GLI repression and accompanying HDAC activity (*Figure 7B, C*). This chromatin-based mode of regulation enables the dynamic control of a field of cells containing primed enhancers. To determine if this priming event occurs on an *ad hoc* basis by disparate inputs or if it is the result of coordinated, HH-independent signaling events, we examined HH-responsive GBRs for the enrichment of additional binding motifs. Besides the GLI motif itself, no other motifs are enriched at high levels (*Figure 1—source data 4*) suggesting that HH-responsive GBRs are a heterogenous population of enhancers with no predominant co-regulators.

Despite being critical for the transcriptional regulation of HH targets, HH-responsive enhancers are a distinct minority, constituting 6% (349/6064) of all GLI-bound, active enhancers. The rest are Stable GBRs with an unclear role in HH transcriptional regulation. Although these enhancers do not have significantly reduced levels of H3K27 enrichment in *Shh*[-/-] limbs, some of them show a trend toward reduced H3K27ac that suggests a continuum of GLI-bound enhancers that range from completely HH-responsive (HH-dependent) to those Stable GBRs that have no HH response

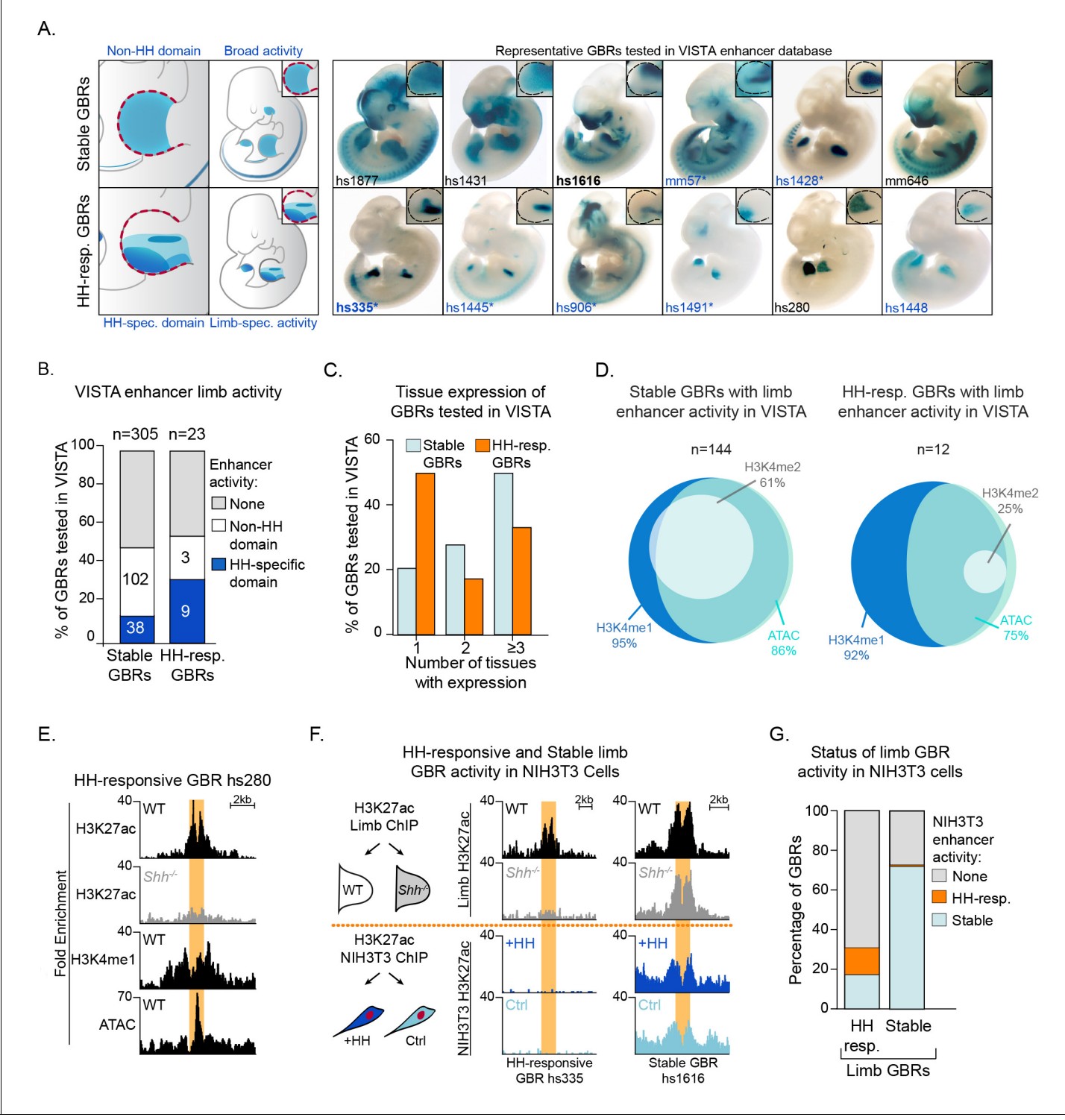

**Figure 6.** Hedgehog-responsive GBRs have tissue-specific enhancer activity within HH-specific domains. (**A**) Enhancers with annotated limb activity in VISTA corresponding to representative HH-responsive GBRs (bottom) and Stable GBRs (top) with limbs magnified and outlined in insets. Limb buds containing HH-specific domains of enhancer activity are indicated by an asterisk. (**B**) Chart indicating total number of VISTA enhancers tested for HH-responsive and Stable GBRs, the numbers of enhancers for each category and their limb enhancer activity. (**C**) Chart delineating the percentage of HH-responsive and Stable limb enhancers that drive expression in one or more tissues. (**D**) Venn Diagram of enhancer marks H3K27ac, H3K4me1, H3K4me2 and ATAC, in Stable and HH-responsive GBRs tested in VISTA that drive expression in the limb. GBRs, are by definition are marked by H3K27ac. (**E**) Enrichment of enhancer markers at a representative HH-responsive GBR tested in VISTA (hs280, *Figure 6A*). (**F**) Schematic of NIH3T3 H3K27ac ChIP

*Figure 6 continued on next page*

*Figure 6 continued*

treated with and without the HH agonist purmorphamine (+HH) and the activity of representative HH-responsive and Stable limb GBRs in response to HH activation in limb and NIH3T3 cells (n = 2). (G) Graph indicating how the acetylation status of HH-responsive and Stable limb GBRs responds to HH signaling in HH-responsive NIH3T3 cells. See *Figure 6—source data 1*; *Figure 6—source data 2*.

The online version of this article includes the following source data for figure 6:

**Source data 1.** Stable and HH-responsive GLI binding regions with limb enhancer activity in the VISTA dataset.

**Source data 2.** NIH3T3 H3K27ac ChIP-seq analyzed data and called peaks.

---

(*Figure 7—figure supplement 1*). Consistent with this, Stable GBRs do have a modest overall increase in H3K27ac enrichment in $Shh^{-/-};Gli3^{-/-}$ limbs on a population level, indicating that their H3K27ac levels are regulated by GLI repressor to some extent. On the other hand, these enhancers are enriched at CpG-rich promoters, which are associated with more broadly expressed genes and have minimal enrichment near HH target genes (*Figure 1G,H*). They are also more highly conserved than HH-dependent GBRs (*Figure 1—figure supplement 1E*). In contrast to HH-responsive enhancers, they appear to be active in other cell types and tissues besides the limb (*Figure 6A*, *Figure 7C*). One possibility is that many Stable GBRs do not have a major role in mediating Hedgehog signaling; GLI repressors at these regions are relatively inert. A second possibility is that GLI repression at Stable GBRs mediates subtle changes to acetylation that confer small reductions in transcription that are beyond the limits of our detection. Finally, it is possible that Stable enhancers are globally active, but engage in long-range collaborations with tissue specific HH-responsive enhancers to activate transcription (*Figure 7D*).

Previous modeling has suggested that GLI repressors within an enhancer work cooperatively through multiple GLI sites (*Parker et al., 2011*), providing another mechanism for tuning enhancer response. HH responsive GBRs contain more GLI motifs than Stable GBRs, which may make them more responsive to GLI repression, although in contrast to this model, they have high quality GLI motifs. As many GLI target genes, including *Ptch1* and *Grem1,* are regulated by multiple GLI enhancers (*Vokes et al., 2008*; *Zuniga et al., 2012*; *Li et al., 2014*; *Lopez-Rios et al., 2014*; *Lorberbaum et al., 2016*), this integration likely extends to higher level hubs of enhancer organization. For example, HH-responsive H3K27ac regions that are not bound by GLI cluster near HH-responsive GBRs, as do Stable GBRs suggesting that they may be modified based on proximity to GLI-repressor-HDAC complexes (*Figure 1—figure supplement 1C*).

The majority of HH-responsive GBRs do not have H3K27me3 enrichment even when there is maximal GLI repression (*Figure 2A–D*; *Figure 7A*). This indicates that the Polycomb repressor complex is not involved in mediating most GLI repression, a conclusion that seemingly conflicts with several studies showing direct or indirect roles for PRC2 in repressing HH transcription. However, these studies largely considered the transcriptional activator targets *Ptch1* or *Gli1* or looked at genetic interactions (*Wyngaarden et al., 2011*; *Shi et al., 2014*; *Lorberbaum et al., 2016*; *Shi et al., 2016*; *Deimling et al., 2018*). Consistent with their findings, *Gli1* has high levels of H3K27me3 enrichment in $Shh^{-/-}$ limb buds (*Figure 2B*). Although *Gli1* and *Ptch1* are often examined in the context of GLI de-repression, they are both GLI-activator genes in that they require the loss of GLI repression as well as subsequent GLI activation for their expression (*Litingtung et al., 2002*; *te Welscher et al., 2002*). GLI activator targets such as these are likely to differ fundamentally in their mode of regulation from those that are activated upon de-repression. As H3K27me3 enrichment is commonly found at promoters (*Young et al., 2011*), GLI repressors on distal enhancers not directly enriched by H3K27me3 might still facilitate the recruitment of PRC2 to promoters through enhancer-promoter interactions. However, only one third of HH target genes have H3K27me3 enrichment at their promoters (*Figure 2—figure supplement 1*; *Figure 2—source data 2*), arguing against this scenario. Additionally, 65% (20/31) of HH-responsive GBRs enriched for H3K27me3 in $Shh^{-/-}$ limbs were detected in the H3K27ac MicroChIP on $Shh^{-/-};Gli3^{-/-}$ limbs. 17/20 of these regions maintained or increased H3K27ac enrichment in $Shh^{-/-};Gli3^{-/-}$ limbs, while the three regions that were reduced were near the GLI-activator-dependent HH pathway genes *Gli1*, *Ptch1* and *Ptch2* (*Figure 4B*; *Figure 4—source data 2*). Thus, for rare limb GBRs requiring GLI activation, their mode of action is consistent with previously proposed models in which GLI activators recruit a complex to remove H3K27Me3,

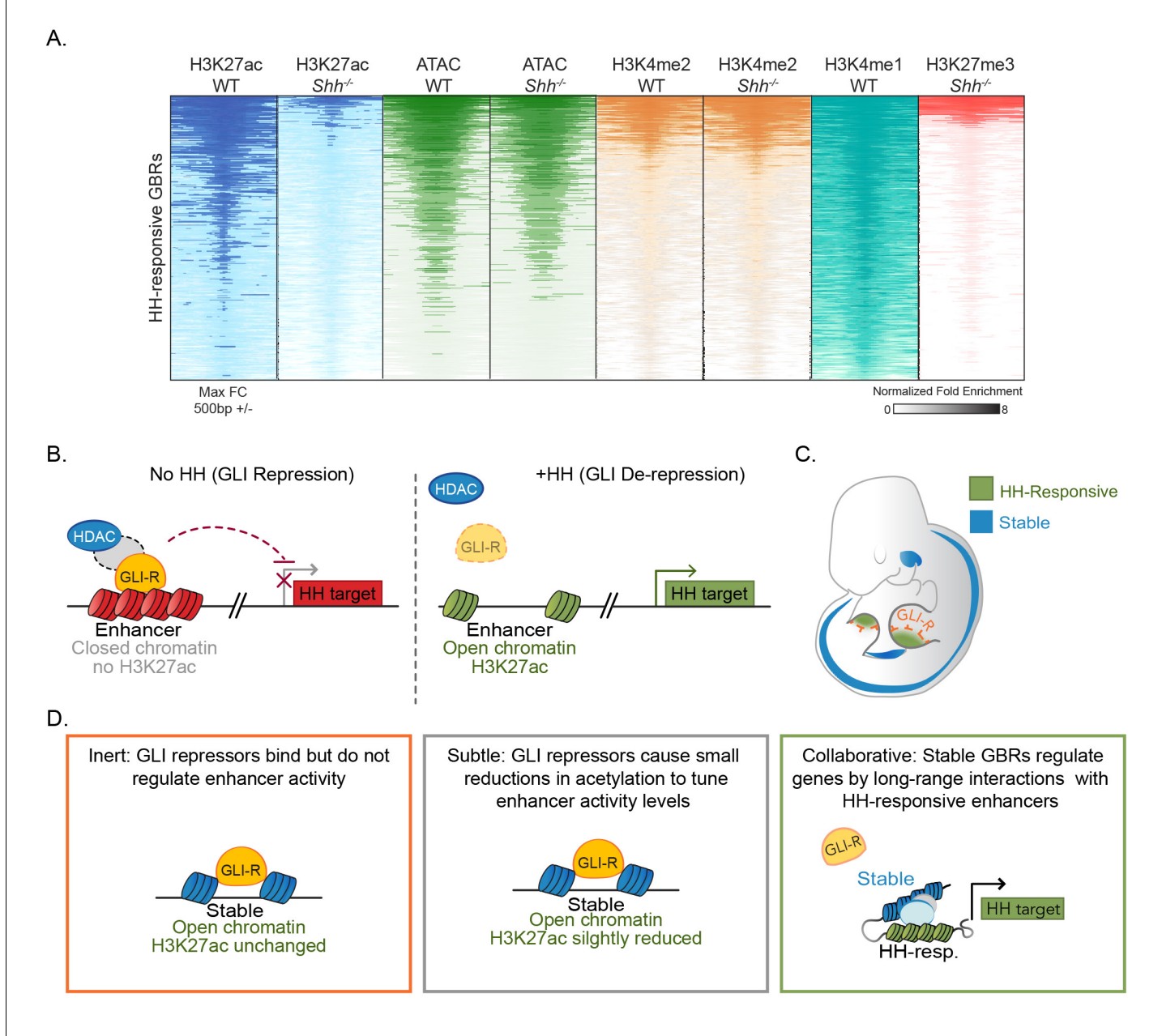

**Figure 7.** Model for GLI transcriptional repression. (A) Summary of enhancer status at HH-responsive GBRs. (B) In the absence of HH, GLI repressors bind to enhancers for HH target genes, limiting their accessibility and, directly or indirectly, recruiting an HDAC complex that de-acetylates Histone H3K27, inactivating the enhancer. In the presence of HH signaling, GLI de-repression and loss of associated HDAC activity result in increased accessibility, the accumulation of H3K27ac and gene transcription. (C) Schematic showing tissue-restricted activity of HH-responsive GBRs within HH-responsive gene expression domains. (D) Possible roles for Stable GBRs in HH transcriptional regulation.

The online version of this article includes the following figure supplement(s) for figure 7:

**Figure supplement 1.** Summary of enhancer status at Stable GBRs.

resulting in the activation of these enhancers and subsequently their cognate target genes (*Shi et al., 2014*).

Confusingly, HDACs have been shown to have properties both consistent with and contradictory to our model. HDACs bind to and deacetylate GLI1 and GLI2 proteins, promoting their ability to act as transcriptional activators (*Canettieri et al., 2010*; *Coni et al., 2013*; *Mirza et al., 2019*). HDACs have also been shown to bind cis-regulatory regions in *Gli1*, consistent with an additional role in

positively regulating HH-mediated transcription (*Zhan et al., 2011*). On the other hand, a SKI-HDAC complex has been shown to bind to and interact genetically with GLI3 to repress anterior digit formation in the limb bud (*Dai et al., 2002*). Similarly, Atrophin acts as a GLI co-repressor by recruiting an HDAC complex (*Zhang et al., 2013*). Multiple studies with SWI/SNF BAF complex members also indicate that they regulate aspects of both GLI activation and repression, roles that have in some cases been shown to be directed by the dynamic association of BAF members with HDAC complexes (*Jagani et al., 2010*; *Zhan et al., 2011*; *Jeon and Seong, 2016*). Our results indicate that HDAC1 is bound to about half of all HH-responsive GBRs. The absence of HDAC1 at such a significant percentage of GBRs could possibly be explained by transient binding of HDACs or the presence of partially redundant HDAC proteins. In support of the latter scenario, HDAC2 has been shown to preferentially bind to distal, rather than promoter regions (*Wang et al., 2009*). Although the simplest model is consistent with GLI repressors directly (via a GLI3 and HDAC-containing repression complex), we cannot exclude the possibility that HDAC1 is constitutively bound at CRMs in a GLI-independent fashion and the HDAC activity occurs indirectly . Collectively, these studies highlight the complexity of GLI regulation and the need for further studies to determine which complexes directly impact GLI repression.

# Materials and methods

**Key resources table**

| Reagent type (species) or resource | Designation | Source or reference | Identifiers | Additional information |
|---|---|---|---|---|
| Genetic reagent (*M. musculus*) | *Gli3^Xt-J Gli3^+/-* | Jackson Laboratory | Jackson Cat# 000026, MGI Cat# 2169581, RRID:MGI:2169581 | Obtained from the Laboratory of Dr. Andy McMahon |
| Genetic reagent (*M. musculus*) | *Shh^tm1amc Shh^+/-* | Jackson Laboratory | Jackson Cat# 003318, MGI Cat# 3584154, RRID:MGI:3584154 | Obtained from the Laboratory of Dr. Andy McMahon |
| Genetic reagent (*M. musculus*) | *Gli3^FLAG* | Laboratory of Dr. Andy McMahon | | Obtained from the Laboratory of Dr. Andy McMahon |
| Genetic reagent (*M. musculus*) | Swiss Webster Wildtype | Charles River | Charles River Cat# NCI 551 IMSR Cat# TAC:sw, RRID:IMSR_TAC:sw | |
| Cell line | NIH 3T3 | ATCC | Cat# CRL-6442, RRID:CVCL_0594 | Used for conventional ChIP-seq |
| Antibody | Anti-H3K27ac (mouse mono-clonal) | Diagenode | Diagenode Cat# C15200184, RRID:AB_2713908 | Used for conventional ChIP-seq |
| Antibody | Anti-H3K27ac (rabbit polyclonal) | Abcam | Abcam Cat# ab4729, RRID:AB_2118291 | Used for conventional ChIP-qPCRs |
| Antibody | Anti-H3K27ac (rabbit polyclonal) | Diagenode | Diagenode Cat# C15410196, RRID:AB_2637079 | Used for conventional MicroChIP-seq |
| Antibody | Anti-H3K27me3 (rabbit polyclonal) | Abcam | Abcam Cat# Ab195477, RRID:AB_2819023 | Used for conventional ChIP-seq |
| Antibody | Anti-H3K4me1 (rabbit polyclonal) | Millipore | Millipore Cat# 07–436, RRID:AB_310614 | Used for conventional ChIP-qPCRs |
| Antibody | Anti-H3K4me2 (rabbit polyclonal) | Millipore | Millipore Cat# 07–030, RRID:AB_11213050 | Used for conventional ChIP-seq |
| Antibody | Anti-M2 FLAG (mouse monoclonal) | Sigma | Sigma-Aldrich Cat# F3165, RRID:AB_259529 | Used for conventional ChIP-seq and WB (1:4000) |

*Continued on next page*

Continued

| Reagent type (species) or resource | Designation | Source or reference | Identifiers | Additional information |
|---|---|---|---|---|
| Antibody | Anti-HDAC1 (rabbit polyclonal | Abcam | Abcam Cat# ab7028, RRID:AB_305705 | Used for conventional ChIP-seq |
| Antibody | Anti-Histone H3 (rabbit polyclonal | Cell Signaling Technology | Cell Signaling Technology Cat# 4499, RRID:AB_10544537 | Used for WB (1:4000) |
| Antibody | Anti-GAPDH (rabbit polyclonal | Cell Signaling Technology | Cell Signaling Technology Cat# 5174, RRID:AB_10622025 | Used for WB (1:1000) |
| Antibody | Anti-B-actin (rabbit polyclonal | Cell Signaling Technology | Cell Signaling Technology Cat# 8457, RRID:AB_10950489 | Used for WB (1:2000) |
| Antibody | Donkey-anti-mouse | Jackson Immuno-Research | Jackson Immuno Research Labs Cat# 715-035-150, RRID:AB_2340770 | Used for WB (1:5000) |
| Antibody | Donkey-anti-rabbit | Jackson Immuno-Research | Jackson Immuno Research Labs Cat# 711-005-152, RRID:AB_2340585 | Used for WB (1:5000) |
| Antibody | Dynabeads M-280 Sheep Anti-Mouse IgG | Invitrogen, Thermo Fisher Scientific | Thermo Fisher Scientific Cat# 11201D, RRID:AB_2783640 | |
| Antibody | Dynabeads M-280 Sheep Anti-Rabbit IgG | Invitrogen, Thermo Fisher Scientific | Thermo Fisher Scientific Cat# 11203D, RRID:AB_2783009 | |
| Chemical compound, drug | Purmorphamine | Stemgent | Stemgent Cat# 04–0009 | Used in cell culture (400 nM) |
| Chemical compound, drug | SAHA | Selleckchem | Selleckchem Cat# MK0683 | Used in limb bud culture (20 µM) |
| Chemical compound, drug | FK228 | Selleckchem | Selleckchem Cat# S3020 | Used in limb bud culture (250 nM) |
| Commercial Assay or Reagent | SensiFAST SYBR-LoROX | Bioline | Bioline Cat# BIO-94020 | |
| Commercial Assay or Reagent | NEBNext DNA Library Prep Master Mix Set for Illumina | New England Biolabs | NEB Cat# E6040L, E7645L | |
| Commercial Assay or Reagent | Agencourt AMPure XP | Beckman Coulter | Beckman Coulter Cat# A63881 | |
| Commercial Assay or Reagent | True MicroChIP Kit | Diagenode | Diagenode Cat# C01010130 | |
| Commercial Assay or Reagent | MicroPlex Library Prep Kit | Diagenode | Diagenode Cat# C05010012 | |
| Commercial Assay or Reagent | Liberase | Roche | Roche Cat# 05401119001 | Cell dissociation (100 µg/mL) |
| Software, Tools | MACS version 2.1.0 | (Zhang et al., 2008 https://github.com/taoliu/MACS | MACS, RRID:SCR_013291 | |
| Software, Tools | limma | (Ritchie et al., 2015) | LIMMA, RRID:SCR_010943 | http://bioconductor.org/packages/release/bioc/html/limma.html |
| Software, Tools | R statistical software | (R Development Core Team, 2014) | R Project for Statistical Computing, RRID:SCR_001905 | https://www.r-project.org/ |
| Software, Tools | CisGenome | (Ji et al., 2008) | CisGenome, RRID:SCR_001558 | http://www.biostat.jhsph.edu/~hji/cisgenome/ |

Continued

| Reagent type (species) or resource | Designation | Source or reference | Identifiers | Additional information |
|---|---|---|---|---|
| Database, Tools | JASPAR motif database | (*Khan et al., 2018*) | JASPAR, RRID:SCR_003030 | http://jaspar.genereg.net/ |
| Database, Tools | Transfac motif database | (*Matys et al., 2006*) | TRANSFAC, RRID:SCR_005620 | http://gene-regulation.com/pub/databases.html |
| Database, Tools | VISTA enhancer browser | (*Visel et al., 2007*) | VISTA Enhancer Browser, RRID:SCR_007973 | https://enhancer.lbl.gov/ |

## Embryonic manipulations

Experiments involving mice were approved by the Institutional Animal Care and Use Committee at the University of Texas at Austin (protocol AUP-2016–00255). The $Gli3^{Xt-J}$ and $Shh^{tm1amc}$ null alleles have been described previously (*Hui and Joyner, 1993*; *Dassule et al., 2000*) and were maintained on a Swiss Webster background. The $Gli3^{3XFLAG}$ allele, with an N-terminal 3XFLAG-epitope, (*Lopez-Rios et al., 2014*; *Lorberbaum et al., 2016*) was maintained on a mixed background. For ChIP and ChIP-seq experiments, fresh E10.5 (32–35 somite) forelimb buds were pooled from multiple litters to obtain sufficient $Gli3^{-/-}$ and $Shh^{-/-}$ mutant embryos along with somite matched controls (Swiss Webster embryos for $Gli3^{-/-}$ experiments and a mixture of WT and heterozygous littermates for $Shh^{-/-}$) embryos. For ATAC-seq, fresh pairs E10.5 (35 somite) posterior forelimb buds were dissected from individual embryos.

To inhibit HDAC1/2, E10.5 embryos (32–35S) were dissected in warm limb bud culture media (*Panman et al., 2006*) and explants still attached to the body wall were cultured in 250 nM of HDAC inhibitor FK228 (Selleckchem S3020), 20 µM of the HDAC inhibitor SAHA (Selleckchem MK0683), or DMSO vehicle control, for two hours at 37C. For each condition, 20–25 embryos were used (n = 4). After incubation, the explants were changed into fresh media (without inhibitor) to dissect anterior limb buds. Cells from anterior limbs were then dissociated and processed for ChIP.

## Cell culture

NIH3T3 were authenticated by and purchased from ATCC (NIH3T3 CRL-1658). They have tested negative for Mycoplasma. Cells were seeded on 6 cm plates with $5 \times 10^5$ cells and grown for three days until completely confluent. They were then switched to low serum (0.5%) and treated with 400 nM purmorphamine (Stemgent 04–0009) or 0.01% DMSO (vehicle control) for 2 days. Under these conditions, a representative purmorphamine-treated sample had substantial elevation of the canonical HH target genes *Ptch1* and *Gli1* compared to controls (47-fold and 697-fold enrichment, respectively). NIH3T3 cells (ATCC CRL-1658) were authenticated and purchased directly by vendor, and tested negative for Mycoplasma.

## Western blots

Whole limb buds from a single litter were lysed for 1 hr at 4C. For fractionation, 500,000 cells from limb buds were then dissociated with 100 ug/mL Liberase (Roche 05401119001), resuspended in CSKT buffer (10 mM PIPES pH6.8, 100 mM NaCl, 300 mM sucrose, 3 mM MgCl$_2$, 1 mM EDTA, 1 mM DTT, 0.5% TritonX-100, incubated on ice for 10 min, and centrifuged for 5 min @ 5000 g. The cytoplasmic fraction (supernatant) and nuclear pellet were each resuspended in loading dye and boiled. Western blots were incubated with the following primary antibodies for 1 hr at room temperature in 3% milk: 1:4000 M2 Flag (Sigma 3165),1:4000 H3 (Cell Signaling 4499), 1:1000 GAPDH (Cell Signaling 5174), 1:1000 H3K27ac (Abcam Ab4729), 1:2000 B-actin (Cell Signaling 8457). Secondary antibodies were incubated for 1 hr at room temperature in 3% milk: 1:5000 Donkey anti-mouse (Jackson 715-035-150), Donkey anti-rabbit (Jackson 711-005-0152).

## Chromatin immunoprecipitation

ChIP experiments were performed as previously described (*Vokes et al., 2008*) with the following modifications. Histone ChIPs were performed on whole E10.5 (32S-35S) forelimbs pooled from 6 to 8 embryos. The GLI3-FLAG ChIP and the H3K27ac ChIP on cultured and treated limbs were

performed on E10.5 (32–35S) forelimbs from 20 to 25 pooled embryos. The HDAC1 ChIP was performed on pooled forelimbs and hindlimbs from 30 E11.5 (40–44S) embryos. Cells were dissociated with 100 ug/ml Liberase (Roche 05401119001) and fixed: 15 min for H3K27ac. 30 min for GLI3-FLAG and 7 min for HDAC1 at room temperature in 1% formaldehyde. After cell lysis, chromatin was sheared. H3K27ac ChIP samples were sheared in buffer containing 0.25% SDS with a Covaris S2 focused ultrasonicator using the following settings: Duty Cycle: 2%, Intensity: 3, Cycles/burst: 200, Cycle time: 60 s, Power mode: frequency sweeping. GLI3-FLAG ChIP samples were sheared using a Branson Sonifier for 10 cycles, 30 s on/60 s off, intensity 3.5. HDAC1 samples were sheared using a Diagenode Bioruptor for 5, 10 min cycles: 30 s on/60 s off, on high power. Sheared chromatin was then split into 3 ChIP reactions and incubated with antibody-dynabead preparations overnight. The H3K27ac antibodies for conventional ChIP were from Diagenode (C15200184) and Abcam (ab4729), while the H3K27Ac antibody for MicroChIPs was from Diagenode (C15410196). Additional antibodies recognized H3K4me1 (Millipore ABE1353) H3K4me2 (Millipore 07–030) and H3K27me3 (Abcam ab195477), FLAG (Sigma F3165) and HDAC1 (Abcam ab7028). Beads were washed 5 times with RIPA buffer (1% NP40, 0.7% Sodium Deoxycholate, 1 mM EDTA pH8, 50 mM Hepes-KOH pH7.5, 2% w/v Lithium Chloride) and 1 time with 100 mM Tris pH8, 10 mM EDTA, 8.0, 50 mM NaCl and then eluted at 70°C for 15 min. For HDAC1 ChIPs beads were washed twice with low salt buffer (0.1% Deoxycholate, 1% Trition X-100, 1 mM EDTA, 50 mM Hepes-KOH pH 7.5, 150 mM NaCl), once in high salt buffer (0.1% Deoxycholate, 1% Trition X-100, 1 mM EDTA, 50 mM Hepes-KOH pH 7.5, 500 mM NaCl), once in LiCl buffer (250 mM LiCl, 0.5% NP-40, 0.5% Deoxycholate, 1 mM EDTA, 10 mM Tris-HCl pH 8), and 2x washes in TE buffer (10 mM Tris-HCl pH 8, 1 mM EDTA). Crosslinking was reversed overnight at 70°C. Chromatin was purified and concentrated, then subjected to quantitative PCR and/or library preparation and sequencing. Quantitative PCR-based analysis was performed using SensiFAST SYBR-LoROX (Bioline BIO-94020) on a Viia7 system (Applied Biosystems). ChIP regions subsequently tested by qPCR are referred to in the figures by the unique peak ID number (*Figure 1—source data 2*). For each biological replicate, 2–3 technical replicates were performed for each qPCR reaction and the Ct values were averaged. Chromatin enrichment was determined by calculating delta delta Ct method (*Livak and Schmittgen, 2001*) against a control region (C1).

Primers are described below. Primers are identified by their H3K27ac Peak ID. Primers labeled #1–5 are HH-dependent GBRs.

| H3K27ac ID | Primers | GBR coordinate | GBR type | Comments |
|---|---|---|---|---|
| 32467 (#1) | F: ACGCAGGCAGTTCCAATACA<br>R: AGGGACTTCACCCAGTTCCA | Chr2:113640572–113641614 | HH-dep. | GRE1<br>(near *Grem1*) |
| 15198 (#2) | F: CCCTCCATTCTCCCTCCTTA<br>R: GGACCTTTCCGTTGAAGTGA | Chr13:63950822–63952750 | HH-dep. | randomly<br>selected GBR |
| 2666 (#3) | F: CTGGCTCCCAGAATCTCTCA<br>R: TGTGCCCCATCTCTTTCAG | Chr1:155211962–155213426 | HH-dep. | randomly<br>selected GBR |
| 45094 (#4) | F: GGGAGGGGTGAACTTGTCTT<br>R: TGCAAATGAACACACGCATA | Chr5:134073187–134074116 | HH-dep. | randomly<br>selected GBR |
| 20941 (#5) | F: TTCCCAGCTCAAGGTCATGT<br>R: AGGAGGCAATGAAGACACTGG | Chr15:86429678–86430690 | HH-dep. | randomly<br>selected GBR |
| 41492 | F: AGAAGGACTCCTATGTGGGTGA<br>R: ACTGACCTGGGTCATCTTTTCA | NONE | NONE | Beta actin-<br>normalizing target |
| 41492 | F: AGCTAACAGCCTGCCCTCTG<br>R: TTTTCCGGTGGTACCCTACG | NONE | NONE | Beta actin-<br>normalizing<br>target for H3K4me1 |
| NONE (C1) | F: GCCAGAATTCCATCCCACTA<br>R: CCAATAACCTGCCCTGACAT | NONE | NONE | negative<br>normalizing |

Samples were processed for 'MicroChIP' using the Diagenode True MicroChIP kit (Cat #C01010130) with the following modifications. Briefly, individual limb pairs (~100 k cells) of wildtype, *Shh*[-/-] and *Shh*[-/-];*Gli3*[-/-] E10.5 embryos (33–34S) were processed separately by dissociating limb buds with 100 ug/mL Liberase (Roche 05401119001), crosslinked for 10 min, lysed and then sheared. Samples were sheared on a Diagenode BioRuptor for six cycles on high, 30 s on/off and processed

through shearing while genotyping in parallel for $Shh^{-/-};Gli3^{-/-}$ and wildtype littermates ($Shh^{+/+};Gli3^{+/+}$). Sheared chromatin was then incubated with H3K27ac antibody (Diagenode C15410196) overnight and Protein A magnetic beads (Diagenode C03010020) the following day for 2 hr. Chromatin-bound beads were washed, eluted and de-crosslinked and purified using MicroChIP DiaPure columns (Diagenode C03040001).

## ChIP-Seq

The ChIP-seq raw datasets from this study have been deposited in GEO (GSE108880) (see Source Data for *Figures 1–5* for processed ChIP-seq and ATAC-seq data). The H3K4me1 data used in this study (GSE86690) were processed and analyzed as all other ChIP experiments were done, described below. All chromosomal coordinates refer to the mm10 version of the mouse genome.

After ChIP was performed as described above, libraries were generated using the NEBNext Ultra II library preparation kit with 15 cycles of PCR amplification (NEB E7645) or generated using the MicroPlex library prep kit (Diagenode C05010012) and sequenced to a depth of >40 million reads per sample for both ChIP and 'MicroChIP' experiments, using two biological replicates. Peaks were called using CisGenome version 2.1.0 (*Ji et al., 2008*). To identify differentially enriched peaks in the WT and $Shh^{-/-}$ limb buds (or control and purmorphamine-treated NIH3T3 cells), the peaks were merged to determine how many WT, WT input, $Shh^{-/-}$ and $Shh^{-/-}$ input reads overlapped with the peak region. The read numbers were adjusted by library size and log2 transformed after adding a pseudo-count of 1. The differential analysis between WT and WT input used limma (*Ritchie et al., 2015*). The FDR of the differential test was obtained and peaks with FDR < 0.05 are determined as having differential signal between WT and WT input. The same differential analysis procedure was repeated to compare between $Shh^{-/-}$ and $Shh^{-/-}$ input, and between WT and $Shh^{-/-}$. To determine GLI motif quality, de novo motif discovery was performed on the 1000 GBRs with the highest quality using the flexmodule_motif function in CisGenome to identify the GLI motif. The GLI motif was mapped to the mouse genome using the motifmap_matrixscan_genome function in CisGenome software with default parameters.

## ATAC-Seq

Individual pairs of posterior forelimb fractions were dissected from 35 somite wildtype (n = 2) or $Shh^{-/-}$ embryos (n = 2). ATAC used components from the Nextera DNA Library Preparation Kit (Illumina) as described previously (*Buenrostro et al., 2015*) with the following variations. 5,000 cells from each sample were added into each reaction and cells were lysed on ice for 8 min. prior to centrifugation. Libraries were generated using 18 cycles of PCR amplification with NEB high fidelity 2x master mix (New England Biolabs), cleaned up with AMPure XP beads (Beckman Coulter) and sequenced on an Illumina NextSeq 500 using PEx75 to a depth of 30 million reads. Peaks were called using MACS2 with a fixed window size of 200 bp and a q-value cutoff of 0.05. Differential analysis of wildtype versus $Shh^{-/-}$ peak signals was performed essentially as described for ChIP above using limma (*Ritchie et al., 2015*).

## Acknowledgements

We thank Blerta Xhemalce, Samantha Brugmann, Kevin Peterson and Janani Ramachandran for comments on the manuscript. We thank Drs. Ken Zaret, Maki Iwafuchi-Doi, Jongwhan Kim and Cathy Rhee for advice on performing ATAC-seq, Andy McMahon for providing the $Gli3^{Flag}$ mice and Jessica Podnar from the Genomic Sequencing and Analysis Facility at the University of Texas at Austin for technical advice. The Texas Advanced Computing Center (TACC) at The University of Texas at Austin provided computational resources. This work was supported by NIH R01HD073151 (to SAV and HJ), The St. Baldrick's Foundation (to SAV) and F31DE027597 (to RKL).

## Additional information

### Funding

| Funder | Grant reference number | Author |
|---|---|---|
| Eunice Kennedy Shriver National Institute of Child Health and Human Development | R01HD073151 | Hongkai Ji<br>Steven A Vokes |
| National Institute of Dental and Craniofacial Research | F31DE027597 | Rachel K Lex |
| St. Baldrick's Foundation | | Steven A Vokes |
| National Institutes of Health | R01HG009518 | Hongkai Ji |

The funders had no role in study design, data collection and interpretation, or the decision to submit the work for publication.

### Author contributions

Rachel K Lex, Conceptualization, Data curation, Formal analysis, Supervision, Funding acquisition, Validation, Investigation, Visualization, Methodology, Project administration; Zhicheng Ji, Conceptualization, Data curation, Software, Formal analysis, Funding acquisition, Validation, Investigation, Visualization, Methodology; Kristin N Falkenstein, Conceptualization, Data curation, Formal analysis, Validation, Investigation, Visualization, Methodology; Weiqiang Zhou, Conceptualization, Data curation, Software, Formal analysis, Investigation, Visualization, Methodology; Joanna L Henry, Conceptualization, Data curation, Software, Formal analysis, Validation, Investigation, Methodology; Hongkai Ji, Conceptualization, Data curation, Software, Formal analysis, Supervision, Investigation, Visualization, Methodology, Project administration; Steven A Vokes, Conceptualization, Software, Supervision, Funding acquisition, Investigation, Visualization, Methodology, Project administration

### Author ORCIDs

Steven A Vokes (iD) https://orcid.org/0000-0002-1724-0102

### Ethics

Animal experimentation: Experiments in this study involving mice were approved by the Institutional Animal Care and Use Committee at the University of Texas at Austin (protocol AUP-2016-00255).

### Decision letter and Author response

Decision letter https://doi.org/10.7554/eLife.50670.sa1
Author response https://doi.org/10.7554/eLife.50670.sa2

## Additional files

### Supplementary files

• Transparent reporting form

### Data availability

Sequencing data has been deposited in GEO (accession GSE108880).

The following dataset was generated:

| Author(s) | Year | Dataset title | Dataset URL | Database and Identifier |
|---|---|---|---|---|
| Lex RK, Ji Z, Falkenstein KN, Zhou W, Henry JL, Ji H, Vokes, SA | 2020 | GLI transcriptional repression regulates enhancer activity and chromatin accessibility for Hedgehog target genes | https://www.ncbi.nlm.nih.gov/geo/query/acc.cgi?acc=GSE108880 | NCBI Gene Expression Omnibus, GSE108880 |

The following previously published datasets were used:

| Author(s) | Year | Dataset title | Dataset URL | Database and Identifier |
|---|---|---|---|---|
| Lewandowski JP, Du F, Zhang S, Powell MB, Falkenstein KN, Ji H, Vokes SA | 2015 | RNA sequencing of mouse littermate wild-type and Shh null E10.5 forelimbs [Illumina] | https://www.ncbi.nlm.nih.gov/geo/query/acc.cgi?acc=GSE58645 | NCBI Gene Expression Omnibus, GSE58645 |
| ENCODE DCC | 2016 | ChIP-seq from limb (ENCSR238SGC) | https://www.ncbi.nlm.nih.gov/geo/query/acc.cgi?acc=GSE86690 | NCBI Gene Expression Omnibus, GSE86690 |

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
