## [Decision Letter]

**Acceptance summary:**

In this manuscript, Lex et al. investigate chromatin modifications in wildtype and Shh mutant undergoing constitutive Gli repression in limb buds. From these data they propose a model to explain dynamic, tissue-specific derepression of genes. Strengths of the approach include: 1) the potential physiological relevance of using a well-studied in vivo system where there is a clear role for Hh signaling and derepressive mechanisms, 2) using a genome wide approach and 3) using the in vivo data of Hh-responsive enhancers (from VISTA datasets) to show that a subset of the putative enhancers are bona fide. The revised paper makes an important contribution to the literature.

The manuscript investigates an important question about how chromatin modifications solidify the dynamic and quick gene expression changes necessary for development and investigates one of the most complicated mechanisms at play- derepression.

**Decision letter after peer review:**

Thank you for submitting your article "GLI transcriptional repression regulates tissue-specific enhancer activity in response to Hedgehog signaling" for consideration by *eLife*. Your article has been reviewed by two peer reviewers, and the evaluation has been overseen by Marianne Bronner as the Senior and Reviewing Editor. The reviewers have opted to remain anonymous. The reviewers have discussed the reviews with one another and the Reviewing Editor has drafted this decision to help you prepare a revised submission.

Summary:

This manuscript examines how GLI transcriptional repression regulates tissue-specific enhancer activity in response to Hedgehog signaling. The authors investigate chromatin modifications in wildtype and Shh mutant with constitutive Gli repression in the limb buds. From these data they propose a model to explain dynamic, tissue-specific derepression of genes.

Essential revisions:

While the reviewers found the manuscript potentially interesting, they also noted that it was difficult to read and evaluate. In general, the paper needs extensive revision with clear definitions and further support for the conclusions. Overlaying the VISTA enhancer database with the ATAC-seq (and ChIP) data would be important. In addition, you need to provide evidence that the Gli3-HDAC mechanism proposed is direct. As the necessary revisions are quite extensive, we include the full reviews for your information.

Reviewer #1:

The Vokes lab has provided important contributions cementing the finding that a significant part of the transcriptional regulation downstream of *Smo* is via the release of *Gli2*/3-mediated suppression of target gene transcription. To further find mechanistic support, this paper probes Gli Binding Regions (GBRs) and finds that GBRs enriched around Hh target genes can lose acetylation in the absence of Shh due to the presence of the repressor forms of Gli3. Although the mechanisms remain unresolved the results indicate that Histone Deacetylases interact with the repressor form of Gli3.

The initial identification of Gli3 binding regions is in WT, E10.5 limb buds, when "high levels of HH target gene expression are observed". According to the release of inhibition model, under these conditions it would be predicted that the Gli3R is not bound to its targets, and would thus not be ChIP-ed. It would seem to me that differential Gli binding in A vs. P or WT vs. *Shh*^-/-^ (or perhaps even better *Smo*^-/-^) limb buds would yield a better collection of relevant GBDs, that would provide a better correlate for the ChIP-seq that is appropriately performed in WT and *Shh*^-/-^ limb buds. Although some elements of this approach are presented in Figure 4, these results are presented as supporting, and not as driving the question.

Using *Ptch1* as the prototypical target for HH signaling is reasonable but *Gremlin* is more complex, as in the developing neural tube its expression is restricted to the roof plate, and thus at best inhibited by Shh signaling, while in the gut there appears little evidence that it is under Shh control. In general, there is very little resolution whether the observed correlation is general (supported by *Gli1* and *Ptch1*) or limb bud-specific.

Figure 4D-F: I don't see the indicated p-values as described in the legend.

Figure 4G-I: What are the "Selected GBRs", and what genes are they associated with?

Purmorphamine is better described as a *Smo* agonist. I do wander about the choice of Purmorphamine, as much more specific small molecules (e.g. SAG) but in particular *Shh* should be used in these experiments, as *Smo* downstream of *Ptch1/2* not necessarily equates *Ptch1/2* inactivation via *Shh*.

Reviewer #2:

In their manuscript, GLI transcriptional repression regulates tissue-specific enhancer activity in response to Hedgehog signaling, Lex et al. investigate chromatin modifications in wildtype and *Shh* mutant (undergoing constitutive Gli repression) limb buds. From these data they propose a model to explain dynamic, tissue-specific derepression of genes. Strengths of the approach include: 1) the potential physiological relevance of using a well-studied in vivo system where there is a clear role for Hh signaling and derepressive mechanisms, 2) using a fairly agnostic genome wide approach and 3) using the in vivo data of Hh-responsive enhancers (from VISTA datasets) to show that a subset of the putative enhancers may be bona fide. However, it is difficult to evaluate whether the data support the conclusions through much of the manuscript due to the lack of definitions and details.

The manuscript investigates an important question about how chromatin modifications solidify the dynamic and quick gene expression changes necessary for development and investigates one of the most complicated mechanisms at play- derepression. As written, the analysis of the VISTA enhancer database provides the only compelling evidence that the authors are, in fact, identifying relevant enhancers. The demonstration that these sites are sensitive to an HDAC inhibitor is consistent with the model that GliR recruits an HDAC but leaves open the possibility of indirect HDAC recruitment. Thus, the manuscript provides further evidence that enhancer-promoter interactions to induce transcription work as the field understands and the advance is that these data may demonstrate it in an in vivo setting.

1) The authors need to define and prove that the GBRs they are examining are, in fact, enhancers. While this possibility is consistent with the data, as currently stands the manuscript does not provide sufficient definitions and analysis for this determination to be made. For example, as stands it is possible that GBRs represent some other type of element, such as a gene body or the 3' ends of genes, or no element at all. As the authors examine H3K4me1/2, they may have appropriate data available.

2) The authors need to define terms such as "around genes", "in close proximity to the promoters", "significantly clustered around", "selecting regions that would not overlap with promoters" as is standard in the field. Without clear definitions of proximity and how the associated genes are being called, these data are not interpretable/overinterpreted. One example: subsection “HH-responsive GBRs are distal enhancers containing high quality GLI motifs”, as explained this paragraph says that the stable GBRs are more likely than Hh-responsive GBRs to be near promoters; however, this does not mean that the rest of the Hh-responsive GBRs are enhancers – – and by the section title “Hedgehog signaling does not regulate other histone modifications at enhancers”, the authors are clearly referring to these distal elements as enhancers.

3) In the section "Hh signaling does not regulate other histone modifications at enhancers", H3K4me1 should shift to H3K4me2 at the active enhancers. The fact that H3K4me2 does not change from *Shh* mutants to wildtype suggests these may not be active enhancers. In fact H3K4me2 only marked "a subset of Hh-responsive GBRs"- why don't they have H3K4me2? If the thought is that they might represent weak enhancers, they should show bidirectional PolII. Related to this in the subsequent section using ATAC-seq, why are the GBRs less accessible under the wild type "activated" condition?

4) The statement stating that the data suggest "a model in which loss of an HDAC-GLI repressor complex leads to acetylation" is a bit misleading as it sounds like the only possibility. In fact, the data suggest a correlation that would likely be true regardless of mechanism and the subsequent culturing with FK228 are again consistent with the model but do not prove it as the GLI-HDAC interaction need not be direct.

5) The authors should examine and report the ATAC-seq and chromatin data specifically at the enhancers from the VISTA enhancer database and discuss in regards to their model.

---

## [Author Response]

Essential revisions:While the reviewers found the manuscript potentially interesting, they also noted that it was difficult to read and evaluate. In general, the paper needs extensive revision with clear definitions and further support for the conclusions. Overlaying the VISTA enhancer database with the ATAC-seq (and ChIP) data would be important. In addition, you need to provide evidence that the Gli3-HDAC mechanism proposed is direct. As the necessary revisions are quite extensive, we include the full reviews for your information.Reviewer #1:[…] The initial identification of Gli3 binding regions is in WT, E10.5 limb buds, when "high levels of HH target gene expression are observed". According to the release of inhibition model, under these conditions it would be predicted that the Gli3R is not bound to its targets, and would thus not be ChIP-ed. It would seem to me that differential Gli binding in A vs. P or WT vs. Shh^-/-^ (or perhaps even better Smo^-/-^) limb buds would yield a better collection of relevant GBDs, that would provide a better correlate for the ChIP-seq that is appropriately performed in WT and Shh^-/-^ limb buds. Although some elements of this approach are presented in Figure 4, these results are presented as supporting, and not as driving the question.

The best way to address this would be with GLI3-Flag ChIPs in *Shh^-/-^*or *PrxCre;Smo^c/c^* limb buds. However, our current Flag ChIP procedure requires fresh tissue and we need large numbers of synchronously staged embryos for collection (typically 20-25 pairs of limb buds at this stage). The extensive breeding required to generate this synchronous population of *Shh^-/-^;Gli3^Flag^* limb buds for ChIP is not feasible for us with our current procedure (we tried but failed to obtain meaningful enrichment with the Flag antibody using the MicroChIP protocol on single pairs of E10.5 *Gli3^Flag^* limb buds).

As an alternative strategy, we dissected E10.5 *Gli3^Flag^* limb buds into anterior and posterior halves (as schematized in Figure 4 and new Figure 5) and generated nuclear and cytoplasmic fractions. There are minimal levels of GLI3 protein in the posterior limb bud where HH is active, while the anterior limb nuclei nearly exclusively express repressor specific GLI3. We have added these data as Figure 1—figure supplement 1B and have added the following section to the Results:

“Nearly all nuclear GLI3 is present in the anterior half of the limb bud in the repressor form with little or no nuclear GLI3 present in the posterior half (Figure 1—source data 1B). Therefore, the GBRs identified in this study are likely to exclusively represent GLI3 repressor binding regions.”

Using Ptch1 as the prototypical target for HH signaling is reasonable but Gremlin is more complex, as in the developing neural tube its expression is restricted to the roof plate, and thus at best inhibited by Shh signaling, while in the gut there appears little evidence that it is under Shh control. In general, there is very little resolution whether the observed correlation is general (supported by Gli1 and Ptch1) or limb bud-specific.

We agree and have edited Figure 1 to highlight domains that have been previously shown to be limb-specific. We have added tracks showing H3K27ac levels in WT and *Shh* null limb buds at *Ptch1* (Lopez-Rios et al., 2014). We also focus on the limb specific GLI enhancer, GRE1, near *Gremlin* (Li et al., 2014), omitting the broader locus.

Figure 4D-F: I don't see the indicated p-values as described in the legend.

2. We have added the p-values to the legend.

Figure 4GI: What are the "Selected GBRs", and what genes are they associated with?

GBR #1 is GRE1, a limb-specific enhancer previously shown to be associated with *Gremlin* (depicted in Figure 1E). The other 4 regions are randomly selected HH-dependent GBRs that were not selected on the basis of being associated with a particular gene and are not near genes. We have added this information in tabular form to the Materials and methods (subsection “Chromatin Immunoprecipitation”). Please note that the referenced figure panels using these primers are now shown in Figure 4G, H and Figure 5D.

Purmorphamine is better described as a Smo agonist. I do wander about the choice of Purmorphamine, as much more specific small molecules (e.g. SAG) but in particular Shh should be used in these experiments, as Smo downstream of Ptch1/2 not necessarily equates Ptch1/2 inactivation via Shh.

As part of our quality control, we reserved a portion of the NIH3T3 cells prior to harvesting for ChIP-seq experiments and used them for generating cDNA to determine the induction of *Gli1* and *Ptch1*. Compared to vehicle treated control cells, those treated with 400nM of Purmorphamine have 47-fold enrichment of *Ptch1* and 697-fold enrichment of *Gli1*. While we agree that the specificity and quantitative aspects of the response might be improved upon activation with Hedgehog ligand, the robust expression of GLI activator targets under our conditions suggests that there is unlikely to be residual GLI repressor activity. We have added the following information to the Materials and methods section:

“Under these conditions, a representative purmorphamine-treated sample had substantial elevation of the canonical HH target genes *Ptch1* and *Gli1* compared to controls (47-fold and 697-fold enrichment, respectively).”

Reviewer #2:[…] 1) The authors need to define and prove that the GBRs they are examining are, in fact, enhancers. While this possibility is consistent with the data, as currently stands the manuscript does not provide sufficient definitions and analysis for this determination to be made. For example, as stands it is possible that GBRs represent some other type of element, such as a gene body or the 3' ends of genes, or no element at all. As the authors examine H3K4me1/2, they may have appropriate data available.

We agree that this is an important point. As H3K4Me2 marks only a minority of regions (see new Figure 7A and response to #3, below), we reanalyzed published H3K4Me1 ChIP-seq data from E10.5 limb buds from the ENCODE consortium and intersected it with GBRs. The majority of the GBRs (83% of HH-responsive GBRs and

81% of Stable GBRs) overlap with H3K4Me1 peaks in addition to being enriched for H3K27ac. Their dual status as H3K27ac+;H3K4Me1+ suggests that these are enhancers. We have added the following statement to the Results:

“As H3K27ac is not exclusively localized to enhancers, we also examined the enrichment of histone H3K4me1, a general marker of primed and active enhancers, at these GBRs, using publicly available data (Encode 2012) (Figure 1—source data 3). In wildtype limb buds, 82% of HH-responsive GBRs are enriched for H3K4 mono-methylation, supporting that these regions are likely to act as enhancers (HH-sens: 123/148, 83%; HH-dep: 162/201, 81%).”

2) The need to define terms such as "around genes", "in close proximity to the promoters", "significantly clustered around", "selecting regions that would not overlap with promoters" as is standard in the field. Without clear definitions of proximity and how the associated genes are being called, these data are not interpretable/overinterpreted. One example: subsection “HH-responsive GBRs are distal enhancers containing high quality authors GLI motifs”, as explained this paragraph says that the stable GBRs are more likely than Hh-responsive GBRs to be near promoters; however, this does not mean that the rest of the Hh-responsive GBRs are enhancers – and by the section title “Hedgehog signaling does not regulate other histone modifications at enhancers”, the authors are clearly referring to these distal elements as enhancers.

We have extensively revised the entire manuscript to define terminology, including the specific example mentioned above, which now reads:

“Although Stable GBRs are not highly enriched at HH target genes, 62% of them (3,544/5,715) are located in close proximity to the promoters of genes (2kb upstream to 1kb downstream of TSS), compared to 26% (91/349) of HH-responsive GBRs (Figure 1H). Most promoter-associated Stable GBRs (90%; 3,190/3,544) are found at promoters associated with CpG islands (defined as a TSS with a CpG region within 5kb upstream to 2.5kb downstream), a quality typically associated with housekeeping genes, and genes that tend to be more broadly expressed and less tissue-specific (Zhu et al., 2008).”

3) In the section "Hh signaling does not regulate other histone modifications at enhancers", H3K4me1 should shift to H3K4me2 at the active enhancers. The fact that H3K4me2 does not change from Shh mutants to wildtype suggests these may not be active enhancers. In fact H3K4me2 only marked "a subset of Hh-responsive GBRs"- why don't they have H3K4me2? If the thought is that they might represent weak enhancers, they should show bidirectional PolII. Related to this in the subsequent section using ATAC-seq, why are the GBRs less accessible under the wild type "activated" condition?

We are unsure why H3K2Me2 is only enriched at a minority of GBRs (now visualized in new Figure 7A). To address this concern more directly, we visualized the presence of several enhancer markers: ATAC-seq, H3K4Me2, H3K4Me1 in those GBRs that have enhancer activity in transgenic embryos and are, by that criteria, functional enhancers. Over half (61% – 81/144) of the Stable GBRs with enhancer activity in the limb have H3K4Me2 enrichment while 25%

(3/12) have H3K4Me2 enrichment in HH-responsive enhancers. To illustrate these findings, we have added Venn Diagrams showing the overlap of enhancer markers in transgenic embryos that drive limb expression (Figure 6D) and have also added ChIP-seq plots for a representative enhancer (Figure 6E).

The data are also summarized for all GBRs in Figure 7A and Figure 7—figure supplement 1. We have added the following section to the Results:

“While all GBRs tested in the VISTA database with limb activity are by definition enriched for H3K27ac, 91% of HH-responsive GBRs and 95% of Stable GBRs are also enriched for H3K4me1 (Figure 6D). Additionally, all GBRS are enriched for at least two markers of enhancers (H3K27ac, H3K4me1, H3K4me2, ATAC) while most are enriched for 3-4 of these markers (67% HH-responsive GBRs; 93% Stable GBRs) (Figure 6D, E).”

To address the question concerning wild type ATAC-seq accessibility, we compared WT GBRs that have called ATAC-peaks (ATAC+) with WT GBRs that do not have ATAC-peaks (ATAC-) (Figure 3B). We have added this information to the Results:

“To determine if these regions are likely to be enhancers, we analyzed the co-enrichment of the enhancer markers H3K4me1 and H3K4me2 at ATAC accessible (ATAC+) and inaccessible (ATAC-) HH-responsive GBRs. […] These results suggest that most of the ATAC- regions are likely to correspond to real enhancers though at a somewhat reduced frequency compared to ATAC+ regions.”

4) The statement stating that the data suggest "a model in which loss of an HDAC-GLI repressor complex leads to acetylation" is a bit misleading as it sounds like the only possibility. In fact, the data suggest a correlation that would likely be true regardless of mechanism and the subsequent culturing with FK228 are again consistent with the model but do not prove it as the GLI-HDAC interaction need not be direct.

We agree, and to address this we performed ChIP-seq to identify HDAC1 binding regions in E11.5 limb buds (since HDAC1 ChIPs require ~30 million cells from ~30 pairs of E11.5 forelimbs, it was not feasible to use E10.5 forelimbs, which would have required ~300 embryos). A sizable number of HH-responsive GBRs are enriched for HDAC1, which supports the possibility that this interaction might be direct. We have moved the HDAC inhibitor data (with FK228 as we showed previously along with a new data using an additional HDAC inhibitor, SAHA, which shows the same trends) and HDAC 1 ChIP-seq data (summaries and representative plots) to new Figure 5 and describe these findings in the Results:

“The increased enrichment of H3K27ac acetylation in HDAC-inhibited anterior limb buds was comparable to that seen in posterior limb buds (Figure 4G). […] We conclude that GLI repressors regulate H3K27ac levels at HH-responsive GBRs through HDACs (see Discussion).”

We note that many GBRs are not enriched for HDAC1, possibly because it is difficult to identify more transient HDAC1 binding events or because they are not enriched for HDAC1. Even if these regions are enriched for another HDAC, such as HDAC2, it does not ‘prove’ that the interaction is direct and we have modified the model Figure 7B (Figure 6A in the previous version) to signify that the interaction between GLI3 and HDAC is unknown (a gray box with dashed lines between GLI3 and HDAC). We have also added the following to the Discussion:

“Our results indicate that HDAC1 is bound to about half of all HH responsive GBRs. […] Although the simplest model is consistent with GLI repressors directly (via a GLI3 and HDAC-containing repression complex), we cannot exclude the possibility that HDAC1 is constitutively bound CRMs in a GLI-independent fashion and the HDAC activity is indirect.”

5) The authors should examine and report the ATAC-seq and chromatin data specifically at the enhancers from the VISTA enhancer database and discuss in regards to their model.

We performed this analysis and include it as Venn Diagrams (Figure 6D). We have included these data in the Results and find that all VISTA GBRs are enriched for at least one other enhancer marker besides H3K27ac (see response to point #3 above) while most are marked multiple markers. We have added the following to the Discussion:

“We find that a subset of GLI-bound regions has chromatin modifications that change in response to HH signaling. […] However, compared to WT embryos, these regions have reduced or absent levels of histone H3K27 acetylation in *Shh*^-/-^ embryos, suggesting a loss of enhancer activity.”

In addition, we depict ATAC-seq and the other enhancer markers examined in this study as Figure 7A and Figure 7—figure supplement 1, which is directly above our model (Figure 7B), which depicts these regions as ‘Enhancer.’